# Enhancing aviation safety: An 80-year data-driven model for classification of aviation incident and accident

Sawera Qureshi[1], Iftikhar Aslam Tayubi[2], Omar BaruKab[2], Sher Afzal Khan[1]*

**1** Department of Computer Science, Abdul Wali Khan University Mardan, Pakistan, **2** Faculty of Computing and Information Technology, King Abdulaziz University, Rabigh, Jeddah, Saudi Arabia

\* sher.afzal@awkum.edu.pk

## Abstract

The aviation system is safety-critical by nature, and any occurrence of an incident or accident can lead to the loss of human life and significant operational disruptions. The International Civil Aviation Organization (ICAO) emphasizes that every flight must take off and land safely—a goal achieved over 126,000 times daily. Despite major advancements,mishaps and accidents continue to occur, underscoring the need for robust safety management systems.The accurate classification of aviation occurrences (Incident or Accident) reports is essential for safety management, yet manual review is time-consuming and prone to inconsistency. While incident/accident labels are assigned during reporting, automated classification enables rapid triage, detection of potential mislabeling, and support for severity assessment in high-volume aviation safety operations.To address this,we developed and compared three machine learning classifiers—Multinomial Naive Bayes, Random Forest, and Support Vector Machine—using TF-IDF vectorization on an 80-year dataset of 53,770 aviation occurrence summaries obtained from the Transportation Safety Board of Canada. A two-stage evaluation strategy was employed, consisting of an initial 80/20 train–test split to create an independent test set, followed by 5-fold cross-validation applied exclusively to the training data to ensure robustness and prevent optimistic bias.The Support Vector Machine (SVM) classifier achieved the highest classification performance, attaining an accuracy of 98.06% during 5-fold cross-validation, with consistent results across folds, demonstrating its effectiveness in managing high-dimensional textual data and dataset complexity. The proposed framework provides a robust foundation for automated aviation safety report processing, offering practical value for (1) early triage of safety reports, (2) identification of potentially mislabeled cases requiring expert review, and (3) integration into downstream severity assessment pipelines. This work advances beyond prior classification studies by establishing a benchmark on the largest historical aviation safety dataset while delivering a deployable and operationally relevant framework for real-world safety management

**Data availability statement:** The data underlying the results presented in the study are available from the Transportation Safety Board of Canada (https://www.bst-tsb.gc.ca/eng/stats/aviation/data-5.html).

**Funding:** This work was supported by King Abdulaziz University (DSR), Jeddah, Saudi Arabia, through the Institutional Fund Projects (grant no. IFPHI-360-830-202 to O.B.). The funders had no role in study design, data collection and analysis, decision to publish, or preparation of the manuscript.

**Competing interests:** The authors have declared that no competing interests exist.

applications. The findings offer valuable insights for regulatory authorities and airline operators, contributing to enhanced safety oversight, improved response strategies, and safer aviation operations.

## 1 Introduction

Aviation safety is a critical concern and a top priority underscored by the International Civil Aviation Organization (ICAO) [1,2] and the Transportation Safety Board (TSB) [3]. The TSB publishes data from its Aviation Safety Information System (ASIS) on reportable accidents and incidents, collectively referred to as aviation occurrences. This data, gathered during investigations, is used to analyze safety issues and identify risks within the Canadian transportation system. Reporting of these accidents and incidents complies with the Transportation Safety Board regulations. The shared goal of ICAO regulations is to ensure safe take-off and landing of flights over 126,000 times daily. Despite substantial progress, significant improvements in reducing accidents and incidents remain necessary, as both continue to pose challenges, highlighting the need for effective and robust safety management systems.

This research study aims to address a significant gap in the automated classification of aviation occurrences.occurrences, specifically incidents and accidents, that were reported in accordance with the Transportation Safety Board regulations [4] in effect at that time, by using a comprehensive 80-year dataset (1955–2020) of occurrence reports from the TSB [5]. Each record in the dataset includes an 'Occurrence Type' field, which labels the record as either an 'Incident' or an 'Accident' based on the ICAO and TSB definitions. These two categories serve as the classification labels for the binary classification task performed in this study.

Aviation safety remains a top priority, and understanding aviation safety trends requires analyzing historical data. Such data, spanning over 80 years, provides valuable insights into the occurrences, causes, and prevention measures of accidents and incidents, as evidenced by the ICAO's stringent safety standards. Terms such as "occurrence," "accident," and "incident" are fundamental in this context. According to ICAO, occurrences are defined as events affecting or potentially affecting the safety of operations, and the occurrence includes any irregular, unplanned, or non-routine event. An accident involves severe injury or aircraft damage, while an incident affects or could affect the safety of operations [6,7].In this study, these ICAO and TSB definitions are used to distinguish between the two classes—incidents and accidents—that the machine learning models are trained to classify.

While previous studies have explored classification of aviation safety reports Madeira 2021, etal. [8], De Vries 2020, etal. [9] Ahadh 2021, etal. [10], their practical deployment in operational safety management remains limited. Our work addresses this gap by developing a classification system with direct operational applications: **(1) Automated Triage** – enabling safety departments to prioritize high-risk reports among thousands of daily submissions; **(2) Quality Control** – identifying potentially mislabeled reports that may require expert re-evaluation; and **(3) Severity**

**Assessment Foundation** – providing accurate initial classification for downstream risk scoring systems. Unlike prior work focusing on specific incident types or shorter timeframes, we establish a comprehensive benchmark on 80 years of data, demonstrating that simple yet well-tuned classifiers can achieve operational-grade accuracy suitable for integration into existing safety management workflows.

Existing literature has highlighted and explored various aspects of aviation safety management practices and the role of data analysis in enhancing aviation safety, including data collection, analysis, and the implementation and development of safety management systems. Previous studies have emphasized the importance of analyzing data to identify trends, understand causes, and implement preventive measures [9, 11–12].

However, a significant and notable research gap exists in automated classification of aviation occurrences, specifically incidents and accidents that can accurately classify them using historical data.

This research study aims to address this gap by developing a classification model to classify aviation occurrences, particularly incidents and accidents, more accurately. The objective is to mitigate risks and enhance response strategies, ultimately improving aviation safety. By utilizing machine learning techniques, the dataset Occurrence.csv was analyzed using Natural Language Processing (NLP) to interpret textual occurrence reports and their summaries.An initial 80/20 stratified train-test split was employed to create an independent test set for unbiased performance evaluation. Three classifiers—Multinomial Naive Bayes, Random Forest, and Support Vector Machine (SVM)—were evaluated in conjunction with TF-IDF vectorization, and 5-fold cross-validation was applied exclusively to the training data to assess model stability and guide model selection, ensuring robust and reliable performance [13–15]. Despite the numerous studies, there is still a lack of effective models for classifying aviation safety occurrences, specifically incidents and accidents, based on historical data. This gap underscores the need for advanced methodologies to improve classification accuracy and reliability.

This study makes three primary contributions:

1. **Operational Framework:** We develop and validate a machine learning pipeline specifically designed for integration into aviation safety management systems, focusing on practical deployability rather than theoretical novelty alone.

2. **Historical Benchmark:** Using the largest available aviation occurrence dataset (80 years, 53,770 reports), we establish performance baselines that account for temporal variations and reporting practice changes over decades.

3. **Practical Value Demonstration:** We demonstrate how high-accuracy classification enables concrete safety applications including report triage, quality assurance, and severity assessment support—addressing the "why classify if already labeled" question by showing operational efficiency gains.

The scope of this research study focuses on analyzing 80 years of historical data from TSB using machine learning techniques, including Natural Language Processing (NLP) techniques like text preprocessing and various classifiers, to achieve high classification accuracy. The aim is to develop machine learning models that accurately classify aviation incidents and accidents, thereby reducing risks, enhancing safety response tactics, and validating these models' performance to improve their accuracy. The constraints include the quality and completeness of the dataset, the complexity of the data, and the need for thorough validation to ensure the model's reliability.Despite these constraints, this research paper aims to provide a reliable system for classifying aviation occurrences, specifically incidents and accidents, leveraging historical data reports and advanced machine learning techniques [16–18].

By addressing these aspects, this research contributes to the field of aviation safety by providing a robust classification system for historical data. This research study offers valuable insights for regulatory bodies and airlines to enhance safety protocols and measures, contributing to safer skies and ensuring safe travel. The findings are expected to have significant implications for proactive safety management in aviation, as they provide a dependable framework for classifying past occurrences which can be used to predict and prevent future incidents and accidents [19–21].

 

The research paper is arranged as follows: The background literature section presents relevant work on aviation safety prediction. The proposed system section explains the methodology and details of the study. The experimental results section displays the results. The discussion section analyzes the outcomes, and the conclusion section includes a summary of the paper's findings and future work.Fig 1 shows the geographic distribution of reported occurrences with the INTERNATIONAL and NATIONAL regions accounting for the largest proportions."

## 2 Background literature

In aviation safety, research has evolved significantly over the years, with various studies focusing on different aspects for the analysis and classification of incident and accident reports. The study of aviation safety through data-driven models has been explored through a range of analytical techniques. Traditional statistical analyses have provided valuable insights into accident frequencies and trends over time, often using aggregated datasets from governmental aviation authorities. More recently, machine learning approaches have enabled the classification of incident reports and prediction of potential risks based on structured features. For instance, prior work has utilized decision trees, SVMs, and neural networks to evaluate causal factors. In parallel, Natural Language Processing (NLP) methods have facilitated the extraction of latent information from unstructured textual reports, revealing human factors, environmental causes, and operational failures. Despite these advances, limitations exist in the temporal scope, integration of multi-source datasets, and emphasis on real-world deployment.

Our work fills this gap by combining structured and unstructured data from an 80-year global occurrence dataset, applying supervised learning and textual analysis to identify risk conditions with an analytical intent.While previous studies have applied NLP/ML to aviation safety, our work is distinguished by its application to the largest and longest temporal dataset, its rigorous, operationally-focused validation strategy designed to prevent leakage, and its explicit aim to provide

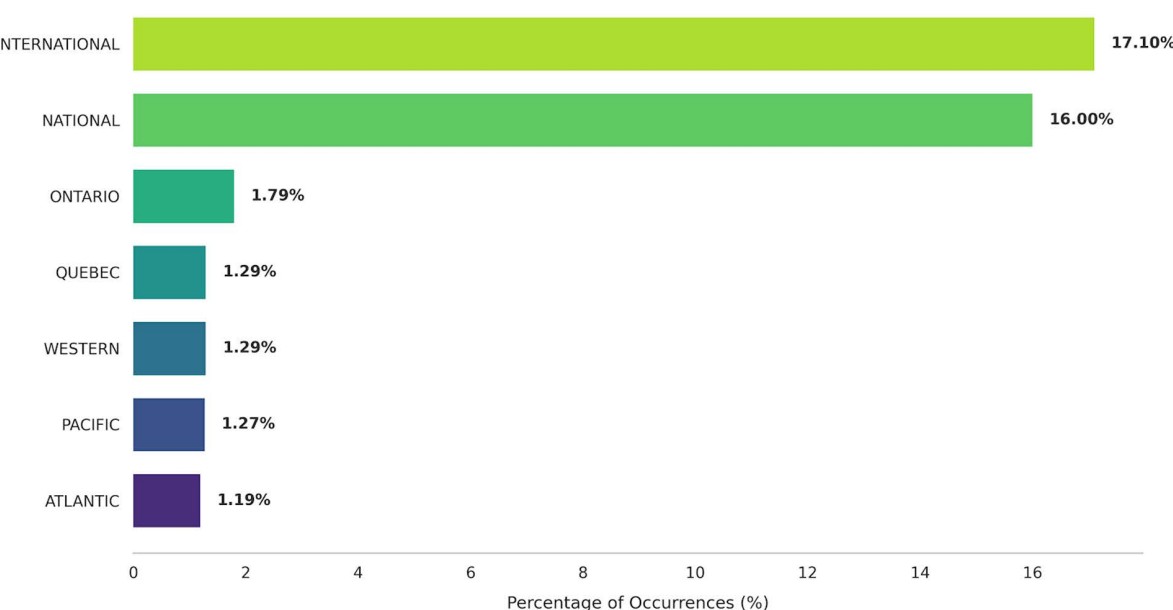

**Fig 1. Regional distribution of aviation safety occurrences (2015-2024).**

a high-accuracy, deployable foundation for automated report triage—a critical first step in high-volume safety management systems.

## 2.1 Previous studies in aviation safety

Lan, H., Wang, S., & Zhang, W. (2024) Investigated the human related maritime accidents types. Using a novel strategi that combine selective ensemble learning and SHAP method the goal is to optimize the accuracy and iterability of prediction system, they contribute by providing a tool to predicting and understanding types of incidents and ensuring a safety measures.

Zhang, X., & Mahadevan, S. (2019) focused on predicting the risk of aviation incidents using ensemble machine learning models. To predict a risk level which are associated with different hazardous causes and their potential consequences. They recognized that traditional methods for analyzing the aviation incidents often they struggled due to these events infrequent and unpredictable nature. They contribute in field of aviation safety by illustrating the worth of machine learning in predicting incident risks and they provide valuable ways for decision-makers.

Jiang, X., Zhang, Y., Li. (2022) Aimed to predict the aircraft passenger satisfaction level and identified the key factors which are influenced. They proposed combining of an RF-RFE-LR model, Random Forest (RF), Recursive Feature Elimination (RFE), and Logistic Regression (LR). they contributed by providing practical approach in the field of aviation by predicting passenger satisfaction and their important key factors for improvement in aviation industry.

Abraham, N., (2022) Used machine learning to categorize FAA unmanned aircraft system (UAS) sighting reports based on their potential hazard levels. They aim to assist prioritizing in aviation authorities that responded to UAS-related incidents effectively.

Madeira et al., (2021) Focusing on predicting human factors involved in aviation incidents using NLP and ML methods. They aimed to identify and classify categories of the human factor from aviation incident reports to improve aviation safety.

de Vries, V., (2020) Explored the application which classified the aviation safety reports with the help of machine learning techniques. They aimed and contribute to categorized the reports based on their content, helping potentially in prevention (safety recommendations), resource allocation and incident analysis.

Ahadh et al., (2021) Proposed a semi-supervised strategy to effectively effectively extract important insights and domain-specific keywords and recognize the underlying topics from accident reports. This approach demonstrates the worth for many applications, Such as risk assessment, accident analysis, and safety improvement measures. This approach combines the topic modeling and keyword extraction to recognize the patterns and key themes with the text data.

Perboli et al., (2021) Demonstrates the power of NLP for automating human factor identification in aviation accidents. By lining up the results with the SHEL (Hardware, Software, Liveware, Environment) methodology, this study provides a structured strategy that make understandable accident causation and make a standard model for accident causation analysis.

Rose et al., (2020) Utilized NLP methods and create a methodology for analyzing aviation safety narratives and recognize the clusters of aviation safety narratives very effectively, offering underlying trends, patterns and by clustering related incidents together these narratives offer possible avenues for safety improvement.

Miyamoto et al., (2022) Focuses on utilizing the NLP techniques to recognize the operational inefficiencies by analyzing the safety reports within the industry of aviation and by extracting the information from textual data, this study goal to reveal patterns which contribute to flight cancellations and delays.

Dong et al., (2022) Proposes a deep learning strategy to address the problem of extracting the casual factors and outperforming conventional methods from aviation incident data. This study aims to improve the aviation safety by recognizing the underlying causes of incidents, reusability and reusability creating it applicable to many text analysis tasks in the aviation Field.

  

Rose et al., (2022) Demonstrates the efficiency of STM in revealing the meaningful topics within the data of aviation safety. STM is a text mining method that going beyond typical topic modeling by adding the external information to guide the discovery of latent themes. This study applies the Structural Topic Modeling (STM) to analyze ASD(Aviation Safety Database) by including the external information, the model offers a more detailed understanding of the causes contributing to aviation incidents events. The finding highlights that how of STM may help to the aviation sector with supporting safety analysis and decision-making.

Jiao et al., (2022) focuses on categorizing incident reports related to Chinese civil aviation into distinct categories and determining the root causes of these incidents and this recognized causes can provide valuable insights for safety improvement concerns. This study shows the effectiveness of integrating machine learning and deep learning methods for classifying aviation incident reports. Their goal is to increase aviation safety by identifying the causes contributing to accidents.

Zhang et al., (2021) Investigates the application of sequential deep learning techniques to predict unfavorable aviation events based on accident investigation records from the National Transportation Safety Board (NTSB). The study advances to the field of aviation safety by demonstrating how deep learning may be used to analyze textual data and to predict unfavorable events Table 1. shows summary of previous studies.

**Synthesis and position of current work.** While the reviewed studies demonstrate valuable applications of ML/NLP in aviation safety, they typically focus on specific aspects (e.g., human factors, risk prediction) or use limited datasets. Our work differentiates itself through three key contributions: (1) Scale and Temporal Scope: We utilize the most extensive historical dataset (80 years, 53,771 reports), enabling analysis of long-term trends and robustness testing across eras; (2) Operational Validation Focus: We implement a rigorous two-tier validation strategy (80/20 split + 5-fold CV on training only) specifically designed to prevent data leakage and provide realistic performance estimates for operational deployment; (3) Practical Application Pipeline: We frame the classification task as the first step in an automated safety report triage and quality assurance system, moving beyond academic accuracy to demonstrate practical utility for safety managers.

## 2.2 Research gaps

Previous studies indicate several areas requiring further investigation. While machine learning and NLP have been applied to aviation safety for tasks such as risk level assessment, incident categorization, and human factors analysis [31], these efforts often focus on specific sub-domains or utilize limited, shorter-term datasets.

A significant gap exists in the development of automated systems for the classification of aviation occurrences (incidents vs. accidents) using a comprehensive, long-term historical dataset. Such a dataset is crucial for building models that are robust to variations in reporting standards, terminology, and technology over time. This study addresses this gap by integrating machine learning and NLP techniques to analyze an unprecedented 80-year dataset, offering new insights into the distinguishing patterns between incidents and accidents.

Furthermore, many existing models, while accurate, are not designed or validated for the specific operational application of automated report processing. They often lack the rigorous validation strategy needed to ensure reliable performance in a real-world safety management workflow, where automated triage and quality control of incoming reports are critical needs.

This research advances the field by:

• Providing a benchmark classification study on the most extensive historical aviation safety dataset available.

• Implementing a validation framework specifically designed to prevent data leakage and yield performance estimates relevant for operational deployment.

• Demonstrating how high-accuracy classification enables direct practical applications—such as automated report triage, detection of labeling inconsistencies, and foundation for severity assessment—rather than focusing solely on academic metrics.

**Table 1. Summary of Previous Studies on ML Classifiers and Feature Extraction Methods.**

| Previous Study | ML Classifiers & FEM | Feature Extracted | Accuracy | Data Source |
|---|---|---|---|---|
| Lan, H., Wang, S., & Zhang, W. (2024) [12] | MLR, BP, SVM, KNN, XG-Boost, Cart & BOW | Weather conditions, Environmental factors, Vessel characteristics, Human factors | 87.50% | MSA, MAIB |
| Zhang, X., & Mahadevan, S. (2019) [21] | Hybrid model (SVM, Deep Neural Network), DT (10-cross-validation) & TF-IDF | Flight specifications, pilot expertise, and aircraft type | 81%, 85% | ASRS |
| Jiang, X., Zhang, Y., Li. (2022) [22] | RF-RFE-LR & KNN, Gaussian Naive Bayes, BP neural network | Flight Characteristics, Passenger experiences & Services | 96% | American airline on Kaggle |
| Abraham, N. (2022) [23] | Neural Network (NN), RF, DT, SVM & Natural Language Processing, TFIDF | Textual descriptions of UAS behavior, Incidents, Altitude, distance, time of day, weather conditions | 96% | FAA USA |
| Madeira et al. (2021) [8] | Semi-supervised Label Spreading (LS), Supervised (SVM) & NLP, TF-IDF or word2vec, Doc2vec Bayesian optimization, HFACS | Identify and classify human factors from aviation incident reports. Aircraft, crew, weather conditions, and circumstances surrounding the incident. | 90%, 77.9%, and 87.5% | ASN database |
| de Vries, V. (2020) [9] | NB, SVM, RF & word embeddings, TF-IDF | Description of incidents (aircraft type, crew information, weather conditions, and circumstances leading to the event) | 80–93% | FAA, EASA & ASRS |
| Ahadh et al. (2021) [10] | GuideLDA | Recognize the phase of flight when occurs aviation accident | 77% | ASRS |
| Perboli et al. (2021) [24] | Word2vec and Doc2vec, SHEL, NLP | Human factors, causes of aviation accidents, Aircraft, crew, weather conditions, and circumstances leading to the accident | 88.89% | Deloitte experts' reports |
| Rose et al. (2020) [25] | BoW with TF-IDF, t-SNE and K-Means Clustering | Extract meaningful trends from narratives | 10 imp clusters and 31 sub-clusters | ASRS |
| Miyamoto et al. (2022) [26] | BoW with TF-IDF, t-SNE and K-Means Clustering | Inefficient operational patterns, flight delays and cancellations (from a safety concern) | Method recognized 7 major clusters and a total of 23 sub-clusters resulted in 1.78% | ASRS |
| Dong et al. (2021) [27] | (AWD) Averaged Stochastic Gradient Descent Weight-Dropped, LSTM | Incidents primary factor and multiple contributing factors from 6 most common factors | 82% average accuracy on 6 common factors and about 89% on the 2 most common factors | ASRS |
| Rose et al. (2022) [28] | LDA with STM | Recognized themes in technical datasets | 80% | ASRS and NTSB |
| Jiao et al. (2022) [29] | TF-IDF, Word2vec, and OC-POS with LR, L-SVM, KNN, DT, NB, SVM, RF, AdaBoost, GBoost, and XGBoost | Causes related to Chinese civil aviation incident reports | F1-score is above 0.90 when identifying 25 causes from the target dataset | Chinese accident reports |
| Zhang et al. (2021) [30] | LSTM | Damage and injury level, Automate prognosis of aviation safety accidents | Accuracy 73%, sensitivity and specificity are 75% and 72.14% | NTSB |

By establishing a reliable, automated method for classifying historical safety reports, this work provides the essential foundation upon which future predictive systems for real-time risk identification can be built. The findings contribute to ICAO's safety objectives by enhancing the efficiency and consistency of safety data analysis, which supports more effective safety monitoring and proactive risk mitigation strategies in the aviation industry.

## 3 Methodology

The primary objective of this study is to develop a machine learning-based classification system that can distinguish between aviation accidents and incidents based on textual summaries from occurrence reports. Our methodology is

structured into several key stages: data acquisition, preprocessing, feature extraction, model training, and evaluation as shown in the Fig 2.

The input to our methodology consists of unstructured textual summaries from the Aviation Safety Information System (ASIS) occurrence reports published by the Transportation Safety Board (TSB) of Canada. Only the `Summary` text field was used as the sole input source for all models, with the `OccTypeID_DisplayEng` field serving exclusively as the target label for classification. This approach was deliberately chosen to prevent data leakage. Specifically, we excluded all structured variables—such as those documented in the comprehensive data dictionary by [1] and detailed in Table 2, which describes 40 key variables spanning occurrence details, aircraft specifications, weather conditions, flight phases, and survivability metrics for Canadian aviation incidents (1955–2020)—because many contain *post-hoc* analysis information (e.g., `DamageLevelID`, `TotalFatalCount`, `InjuriesEnum`) that would provide the model with investigation outcomes unavailable at the time of prediction. Crucially, the `Summary` field itself is a synthesized narrative that incorporates the essential informational content from these structured variables. For instance, details on weather (`SkyCondID`), aircraft type (`AircraftModelID`), and flight phase (`PhaseID`) are naturally embedded within the textual report. Our NLP pipeline, employing TF-IDF vectorization, transforms this single but information-rich text column into multiple numerical features, allowing the models to learn patterns from the circumstances described at the time of the event, thereby ensuring a realistic and leakage-free classification framework. The output is a binary classification indicating whether a report corresponds to an accident or an incident, aiming to assist safety authorities in proactive hazard identification and streamlined report processing.

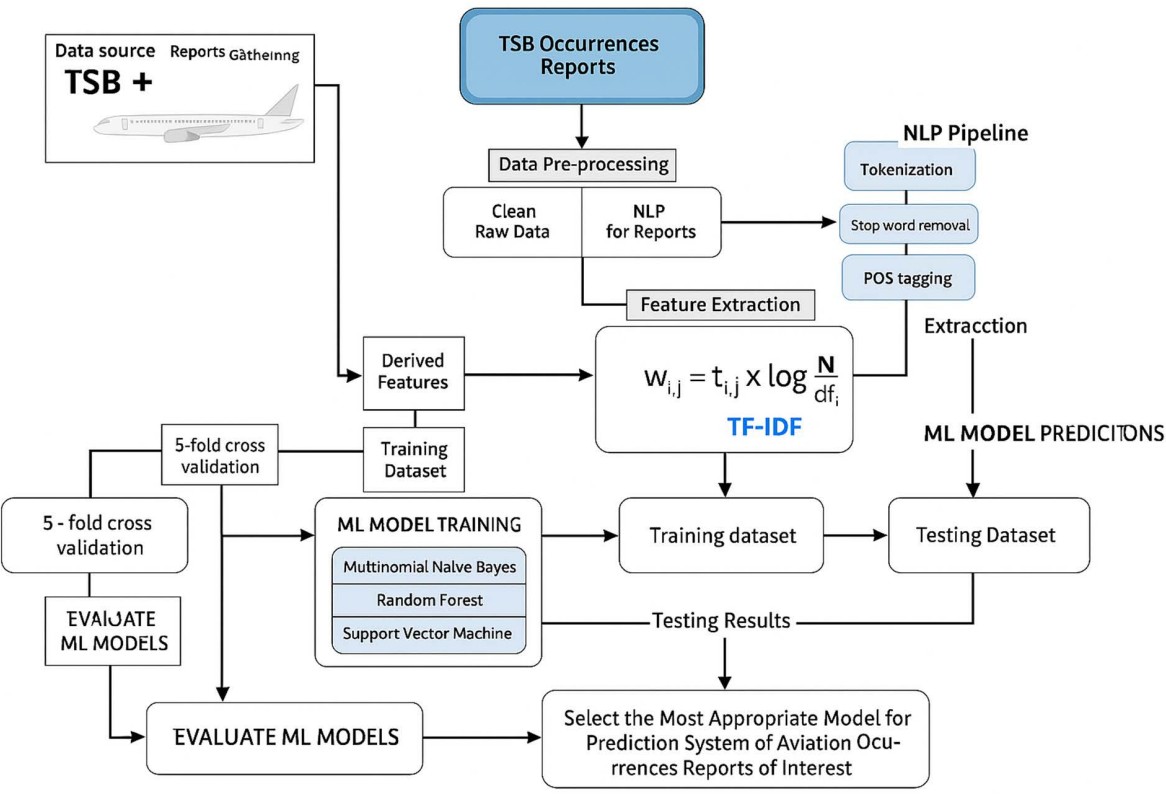

**Fig 2. Architecture of Methodology.**

**Table 2. Data Dictionary of 40 Key Variables for Canadian Aviation Safety Incident Analysis (1955-2020).**

| Feature Name | Description |
|---|---|
| PhaseID | Flight phase during the occurrence (multiple phases possible) |
| **OccTypeID_DisplayEng** | **Description of occurrence type (accident/incident) – Used ONLY as target label** |
| OrganizationID | Operator's organization name |
| SurvivableEnum | Indicates if the occurrence was survivable (Yes/No/Unknown) |
| SurvEquiID | Survival equipment available on the aircraft (multiple entries possible) |
| TimeZoneID | Time zone used for reporting the occurrence |
| LightCondID | Light conditions at the time of occurrence |
| YearOfManuf | Aircraft's manufacturing year |
| AircraftModelID | Aircraft model |
| TotalSeriousCount | Total number of serious injuries (includes ground injuries) |
| InjuriesEnum | Indicates if there were any injuries (includes ground injuries) |
| ICAO | ICAO occurrence category (multiple entries possible) |
| AircraftMakeID | Aircraft make |
| EvacEgressIssueEnum | Indicates if there were evacuation egress issues |
| SkyCondID | Sky conditions at the time of occurrence |
| GeneralWeatherID | Weather conditions conducive to visual or instrument flight rules |
| EquipEffReasonID | Reason for survival equipment effectiveness |
| AircraftEvacTime | Duration of aircraft evacuation (in minutes) |
| SurfaceContaminationID | Type of surface contamination (multiple entries possible) |
| FlightPlanTypeID | Type of flight plan |
| TotalFatalCount | Total number of fatalities (includes ground fatalities) |
| OccTime | Time of the occurrence (24-hour format) |
| VisibilyCeiling | Visibility ceiling (in feet) |
| OccNo | Unique occurrence number |
| OccRegionID | Region of the occurrence |
| OccDate | Date of the occurrence (YYYY-MM-DD format) |
| Visibility | Visibility (in statute miles) |
| AircraftTypeID | Aircraft type as defined by Canadian Aviation Regulations |
| RunwaySurfaceID | Runway surface texture |
| **Summary** | **Summary of the occurrence – Used ONLY as input feature** |
| EvacHamperedID | Reasons for evacuation being hampered |
| EquipEffectiveEnum | Indicates if the equipment was effective |
| DamageLevelID | Aircraft damage level as defined by ICAO |
| WeatherPhenomenaTypeID | Type of weather phenomena at the time of occurrence (multiple entries possible) |
| CountryID | Country of the occurrence |
| OperatorTypeID | Type of operator (private, commercial, state) |
| EquipInfluenceEnum | Indicates if equipment affected survivability (Yes/No/Unknown) |

## 3.1 Algorithm 1: Framework of Our Methodology

**Algorithm 1 Classification of Aviation Occurrence (Incident or Accident)**

```
Input: Raw dataset D from TSB
Output: Classification of aviation occurrence (Incident or Accident)
begin
    Preprocess D (Remove special symbols, stop-words, tokenize text data).
    Label occurrences in D
    (Incident or Accident).
    Extract features from D
    Using TF-IDF vectorization.
    Classify D
    Using MNB, RF, and SVM models
    Optimize performance
    with 5-fold cross-validation.
end
```

**3.2 Dataset description.** The dataset used in this research comprises occurrence reports from the Transportation Safety Board (TSB) of Canada, covering approximately 80 years from 1955 to 2020. We selected the Canadian TSB dataset due to its public availability, consistent structure, and comprehensive documentation across a long historical span. Notably, while it originates from Canada, the dataset captures a diverse range of aviation incidents and accidents that include reports involving international carriers, thus offering insights applicable beyond national boundaries.Some of these reports were also collected from the National Transportation Safety Board (NTSB) of the United States to ensure alignment with ICAO definitions. However, the dataset spans decades and includes reports compiled by different experts and groups, which may introduce potential biases due to variations in reporting standards and practices over time.

The TSB published the data from its Aviation Safety Information System (ASIS), which includes reported accidents and incidents—collectively referred to as occurrences. This data, gathered through official investigations, serves to analyze safety concerns and identify risks within the Canadian and broader aviation systems. These reports comply with the Transportation Safety Board Regulations applicable at the time of each event.The Transportation Safety Board (TSB) Regulations, under the Canadian Transportation Accident Investigation and Safety Board Act, establish mandatory reporting requirements for aviation occurrences. Revised in 2014, the regulations specify incidents to be reported, including structural failures, injuries, and system malfunctions. Reports must include detailed information and be submitted promptly, with exemptions applying only if such information has already been submitted. The TSB retains the authority to request further data to ensure comprehensive investigations and effective safety oversight.

Labels for occurrences were assigned based on definitions provided by the ICAO (the UN agency that sets international civil aviation standards).The dataset maintains its natural class distribution with 30,270 incidents (56.4%) and 23,500 accidents (43.6%). No resampling, weighting, or balancing strategies were applied to preserve the real-world incidence rates and avoid artificial performance inflation. The dataset includes detailed records of incidents and accidents categorized by multiple operational factors and comprises a total of 53,770 samples: 30,270 incidents and 23,500 accidents. The dataset used in our modeling covers air transportation occurrence data from January 1955–2020, encompassing the full 80-year period.

Our focus is on analyzing conditions that lead to aviation incidents and accidents, not on modeling routine, uneventful flights—which are, by default, considered safe. The absence of risk-indicative patterns in a report typically implies normal operations. Thus, by identifying patterns associated with accidents or incidents, the model indirectly aids in recognizing conditions less likely to produce adverse outcomes. The objective is not to predict "safe flights" per se but to highlight high-risk factors that may warrant preventive or mitigative interventions. This dataset is available at:https://www.bst-tsb.gc.ca/eng/stats/aviation/data-5.html

The 80-year span (1955–2020) presents unique challenges not present in shorter-term studies: evolving reporting standards, terminology shifts, changes in aircraft technology, and varying regulatory frameworks. Unlike models trained on short-term data, our model must learn robust patterns that generalize across these temporal variations. This necessitates a validation strategy that tests stability across time—addressed through our two-tier validation approach—and ensures the model does not overfit to era-specific jargon or reporting styles.

**3.2.1 Dataset overview.** *Structured Fields.* The structured data fields in the dataset include:

- **Date and Time:** specific date and time of occurrence took place.

- **Location:** The geographic area where the incident or accident occurred.

- **Occurrence Type:** Classification of the occurrence, such as accident, incident, or serious incident.

- **Occurrence Category:** Further categorization depends on the nature of the event.

- **Aircraft Details:** Information related the aircraft involved in the occurrence, including model and registration.

- **Injuries/Fatalities:** No of fatalities or injuries resulting from the occurrence events.

- **Weather Conditions:** The weather conditions at the time of occurrence.

- **Aerodrome Data:** Information about the landing and takeoff aerodromes or operating surfaces.

*Unstructured textual data.* The dataset occurrence also includes unstructured textual descriptions of each event. These descriptions provide detailed narratives report of the events occurred, which are very important for understanding the context and their contributing factors. Fig 3 shows the distribution of the dataset used in this study, which comprises a total of 53,770 samples across two classes. The dataset is moderately imbalanced, with "INCIDENT" being the most common class (30,270 samples, 56.3%) and "ACCIDENT" being the least common (23,500 samples), resulting in an imbalance ratio of 1.3:1.

**3.2.2 Data analysis techniques.** The study consists on the following data analysis techniques:

**3.2.3 Natural language processing (NLP).** Refers to the method of human communication through text and speech. As a branch of AI, it enables machines to understand and manipulate human languages, facilitating interaction between humans and computers using natural language [32]. It aims to create algorithms and systems capable of understanding and processing both structured and unstructured language data to support decision-making processes. This field has gained significant momentum in recent years, allowing systems to read, decipher, understand, and derive meaningful insights from human language. These capabilities enable the development of systems that can performed the tasks like grammar checking, translation and classification of topics [33].

The companies are increasingly using the natural language processing tools to extract valuable insights from data and they automate the routine tasks [34]. NLP applications include chatbots, Google Assistant, Alexa, Siri, and auto correctors like Grammarly. In aviation, various techniques are employed to process and analyze textual data from occurrence reports. These techniques convert text into a format suitable for analysis, improving the performance of models and enhancing aviation safety. The main steps in NLP include text preprocessing, feature extraction, model training, and model evaluation.

## 3.3 Text-preprocessing

Text preprocessing stage is also known as NLP pipeline it is set of text pre-processing elements which are connected in the series [35,36]. These sequential steps transform raw text into a format that can be understood and processed by a computer. This stage is particularly crucial for predicting aviation incidents and accidents, as it converts unstructured textual reports into a suitable format for machine learning algorithms.

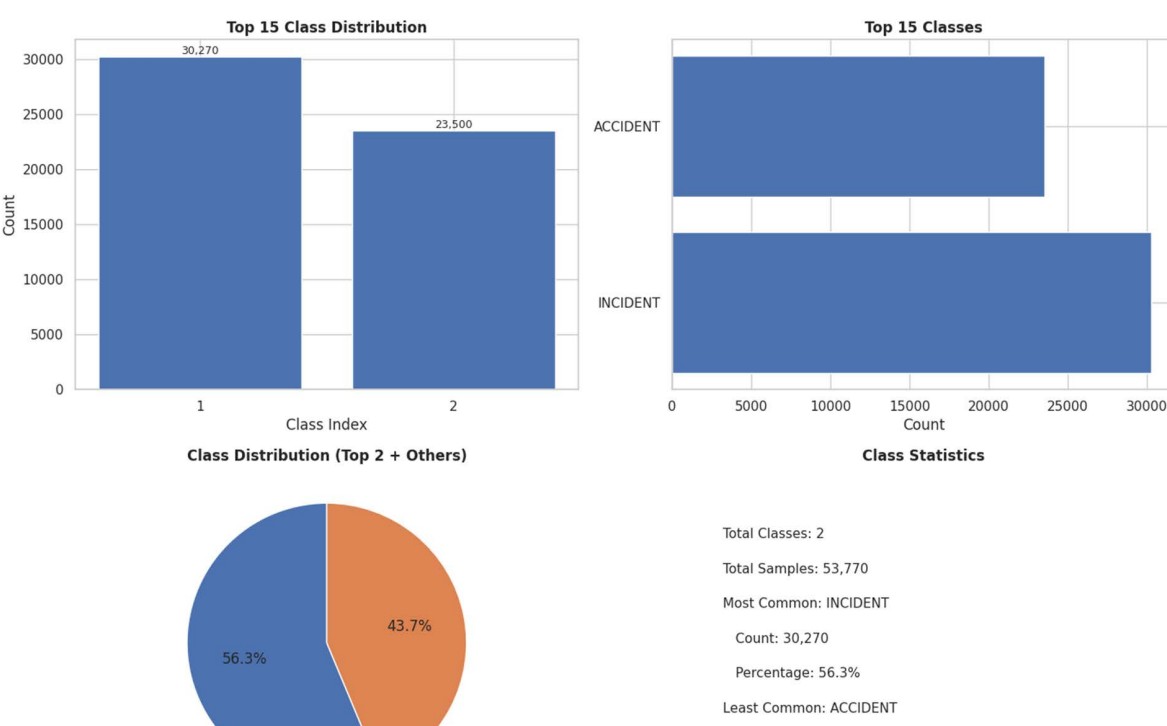

**Fig 3. Dataset Class Distribution Analysis.**

During this stage, irrelevant characters or symbols that could distort the analysis process are removed. For instance, URLs are eliminated as they do not contribute to the contextual background of accidents or incidents. Additionally, punctuation, non-alphabetic or numeric characters, and stop words are removed because they do not provide valuable information for analysis.

The text preprocessing pipeline was implemented in Python using NLTK (v3.8.1) and scikit-learn (v1.3.0), and included the following sequential steps:

**Lowercasing:** All text was converted to lowercase.

**Removal of URLs and Special Characters:** Non-alphanumeric characters (except spaces) and URLs were removed using regular expressions.

**Tokenization:** Text was split into tokens (words) using NLTK's word_tokenize function.

**Stop Word Removal:** We removed standard English stopwords from the NLTK corpus, supplemented with a custom list of 25 aviation-specific stopwords (e.g., 'aircraft', 'pilot', 'runway', 'airport', 'flight') that were overly frequent and non-discriminative. A full list is provided in Supplementary Material S1.

**Lemmatization (Primary):** Tokens were lemmatized to their base dictionary form using the WordNetLemmatizer from NLTK, with part-of-speech tagging where applicable. Stemming was tested but not used in the final pipeline, as lemmatization provided more semantically coherent terms.

**Handling of Numbers & Units:** Numbers were retained as they can indicate severity (e.g., "2 injuries"). Units of measurement (e.g., "feet," "knots") were kept.

**Domain-Specific Standardization:** Common aviation abbreviations (e.g., "VFR" → "visual flight rules," "ATC" → "air traffic control") and French terms in bilingual Canadian reports were standardized to English equivalents.

**Vocabulary Pruning:** After vectorization, terms occurring in fewer than 5 documents (min_df = 5) were excluded to remove noise and extremely rare terms.

In the tokenization process, textual summaries of reports are split into smaller units called tokens, which are easier for machine learning models to process. The lemmatization process is employed to return words to their base form, while the stemming process reduces words to their root form. Both stemming and lemmatization are particularly important for this task because they ensure that different word forms are treated as the same entity (e.g., "fly," "flying," and "flown" are all reduced to "fly"). This enhances the consistency of the textual data, reduces redundancy, and helps the models focus on the core meaning of words rather than their variations.

For this research study, WordNetLemmatizer and Porter Stemmer were used to perform lemmatization and stemming, respectively. These text preprocessing steps ensure that the textual summaries of reports are clean and consistent, enabling machine learning models to predict and analyze aviation incidents and accidents effectively.

Table 3 illustrates some examples of textual data before and after preprocessing from the TSB occurrence reports.

## 3.4 Feature extraction

This method is crucial for deriving valuable features from datasets, as this process plays a pivotal role in text processing. It serves as the foundation for tasks such as classification. It is necessary for researchers to focus on the extraction of the most appropriate information from the raw data [37,38]. In this study, we employ a single feature extraction technique, specifically Term Frequency-Inverse Document Frequency (TF-IDF) vectorization. This method is used to convert and

**Table 3. Comparison of Original and Preprocessed Reports.**

| Original Reports | Preprocessed Reports |
|---|---|
| **Accident:** C-FGWO, a 49 North Helicopters Limited Robinson R22 Beta, departed Tofino / Long Beach Airport (CYAZ), BC, on a training flight under visual flight rules to Campbell River Airport (CYBL), BC. During an off-field landing attempt, the helicopter's low rotor RPM warning horn sounded. The training pilot took control, but the helicopter landed hard and rolled over onto its left side. The helicopter sustained substantial damage, including damage to the airframe, tail boom, and main rotors. The crew telephoned for assistance and were extracted. The helicopter was subsequently airlifted to the company's facility in CYBL. It was determined that the helicopter is beyond viable economic repair and has been written off. | **Accident:** cfgwo 49 north helicopters limited robinson r22 beta departed tofino long beach airport cyaz bc training flight visual flight rules campbell river airport cybl bc off-field landing attempt helicopter low rotor rpm warning horn training pilot took control helicopter landed hard rolled left side helicopter sustained substantial damage airframe tail boom main rotors crew telephoned assistance extracted helicopter airlifted company facility cybl determined helicopter beyond viable economic repair written off |
| **Incident:** C-FEQW, a Summit Air Ltd. Dornier DO228−202, was operating as a training flight SUT01T at the Yellowknife Airport (CYZF), NT. During a crosswind landing, after brake application, the left main landing gear tire burst, resulting in a loss of directional control, and the aircraft veered off the side of the runway. There were no injuries or damage to the aircraft. | **Incident:** cfeqw summit air ltd dornier do228202 operating training flight sut01t yellowknife airport cyzf nt crosswind landing brake application left main landing gear tire burst loss directional control aircraft veered side runway no injuries damage aircraft |

transform textual data into numerical vectors, making it interpretable by the machine. By using this technique, we are able to extract features that help identify unique words across the entire dataset. This technique can improve classification or prediction accuracy and maximize the effectiveness and relevance of the feature extraction process, leading to more meaningful outcomes.

The TF-IDF vectorization was implemented using `TfidfVectorizer` from the scikit-learn library (version 1.3.0) with the following parameters:`max_features = 5000` (to limit dimensionality while retaining informative terms), `ngram_range=(1, 2)` (to capture unigrams and bigrams), and `min_df = 5` (to exclude extremely rare terms). The vocabulary size after transformation was 4,872 features, representing the most discriminative terms for aviation safety reports.

**3.4.1 TF-aIDF.** This statistical method is widely used in NLP for transforming text data to numerical features in information retrieval tasks. This method originates from language modeling theory [13].

In this theory, the words within a text can be categorized into two types: words with eliteness and those without it. Eliteness refers to the importance or significance of certain words in a document or set of documents. This method's calculation involves combining two key metrics: one metric evaluates the frequency of a word within a document, while the other assesses the word's inverse document frequency. In a document, term frequency (TF) measures how often a word appears, whereas inverse document frequency (IDF) measures its importance across a document collection. They aid in distinguishing and classifying documents by assigning importance or weight to words that are unique to a specific set of documents.High- or low-frequency terms are weighted more significantly by IDF.The final TF-IDF score is calculated by multiplying TF and IDF values for a term in a document.

This combination is referred to as TF-IDF. As shown in Eq (1), the mathematical expression for the weight of a term in a document using the TF-IDF method is represented as:

$$W(d, t) = TF(d, t) * \log \left( \frac{N}{df(t)} \right)$$

In this equation, TF(d,t) represents the number of times the term t appears in document d divided by the total number of terms in d, N represents the total number of documents, and df(t) denotes the number of documents that contain the term t in the corpus. The first term enhances recall, while the second improves precision. Although TF-IDF resolves the issue of frequently occurring terms within a document, its score indicates a term's importance in the context of the entire corpus.

The TFIDF vectorization was implemented using `TfidfVectorizer` from scikit-learn with the following parameters: `max_features = 5000`, `ngram_range=(1,2)` (to capture unigrams and bigrams), `min_df = 5, max_df = 0.95, sublinear_tf = False`, and `smooth_idf = True`. The final vocabulary size after applying `min_df` and `max_features` was 4,872 terms.

However, it treats each word as an individual index and does not account for word similarity. This method contributes to dimensionality reduction by selectively emphasizing the most crucial terms, thus focusing on the key aspects of the text data.

Fig 4 illustrates the top 20 most discriminative TF-IDF features (terms) extracted from the aviation safety reports, showing their relative importance for accident vs. incident classification. Notable aviation-specific terms include *crashed*, *substantial damage*, *injuries* (associated with accidents), and *runway excursion*, *loss of control*, *malfunction* (associated with incidents).This analysis confirms that the model learns features directly related to the outcome-based definitions, which is appropriate for the classification task.

The feature importance analysis revealed that TF-IDF effectively captured domain-specific terminology critical for aviation safety classification, with bigrams (e.g.,*substantial damage*, *runway excursion*) providing additional contextual value beyond single words.

In addition to individual feature importance, an overall analysis of the TF-IDF feature space was performed to examine its statistical characteristics. Fig 5 shows that the TF-IDF representation is highly sparse, dominated by aviation-specific

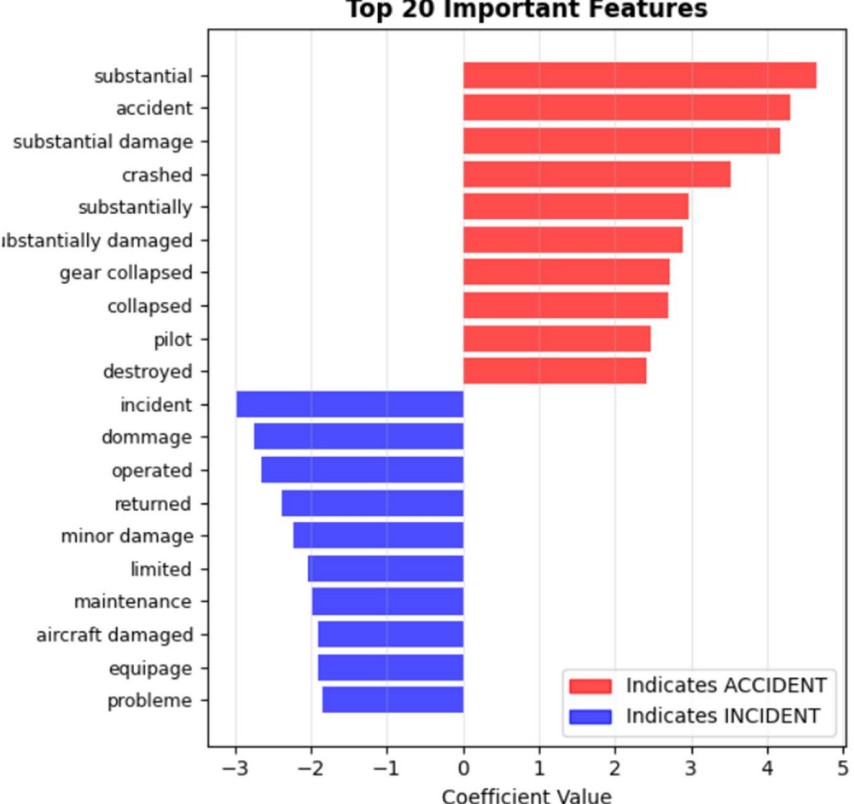

**Fig 4. Top 20 discriminative TF–IDF features for accident vs. incident classification.**

terminology, and exhibits a clear distribution of term weights across documents. The presence of both unigrams and meaningful bigrams indicates that the vectorization captures relevant contextual information, supporting the suitability of TF-IDF features for aviation occurrence classification.

Nevertheless, TF-IDF vectors often yield higher accuracy compared to other techniques [39].

While TF-IDF with standard classifiers is a well-established NLP pipeline, our methodological innovation lies in its application to the largest and longest temporal aviation safety dataset and the rigorous validation strategy designed to ensure operational reliability. Rather than proposing a novel algorithm, we demonstrate that a carefully tuned, simple pipeline can achieve operational-grade accuracy (98.06%) suitable for real-world deployment. Furthermore, our feature extraction includes aviation-specific preprocessing (e.g., standardizing abbreviations like VFR, ATC) and bigram modeling to capture contextual phrases (e.g., "substantial damage," "runway excursion"), which enhances domain relevance beyond generic text processing.

### 3.5 Machine learning models

In this study, ensemble classifiers were trained on the training set to classify occurrence reports in the dataset and evaluated on the test data. The ML algorithms used in this study are Multinomial Naive Bayes, Random Forest, and Support Vector Machine (SVM). These supervised machine learning classifiers belong to the ensemble learning family.These models are chosen for their strong classification capabilities and robustness in handling large, extensive datasets.These classifiers were selected based on their proven effectiveness in text classification tasks, computational efficiency, and

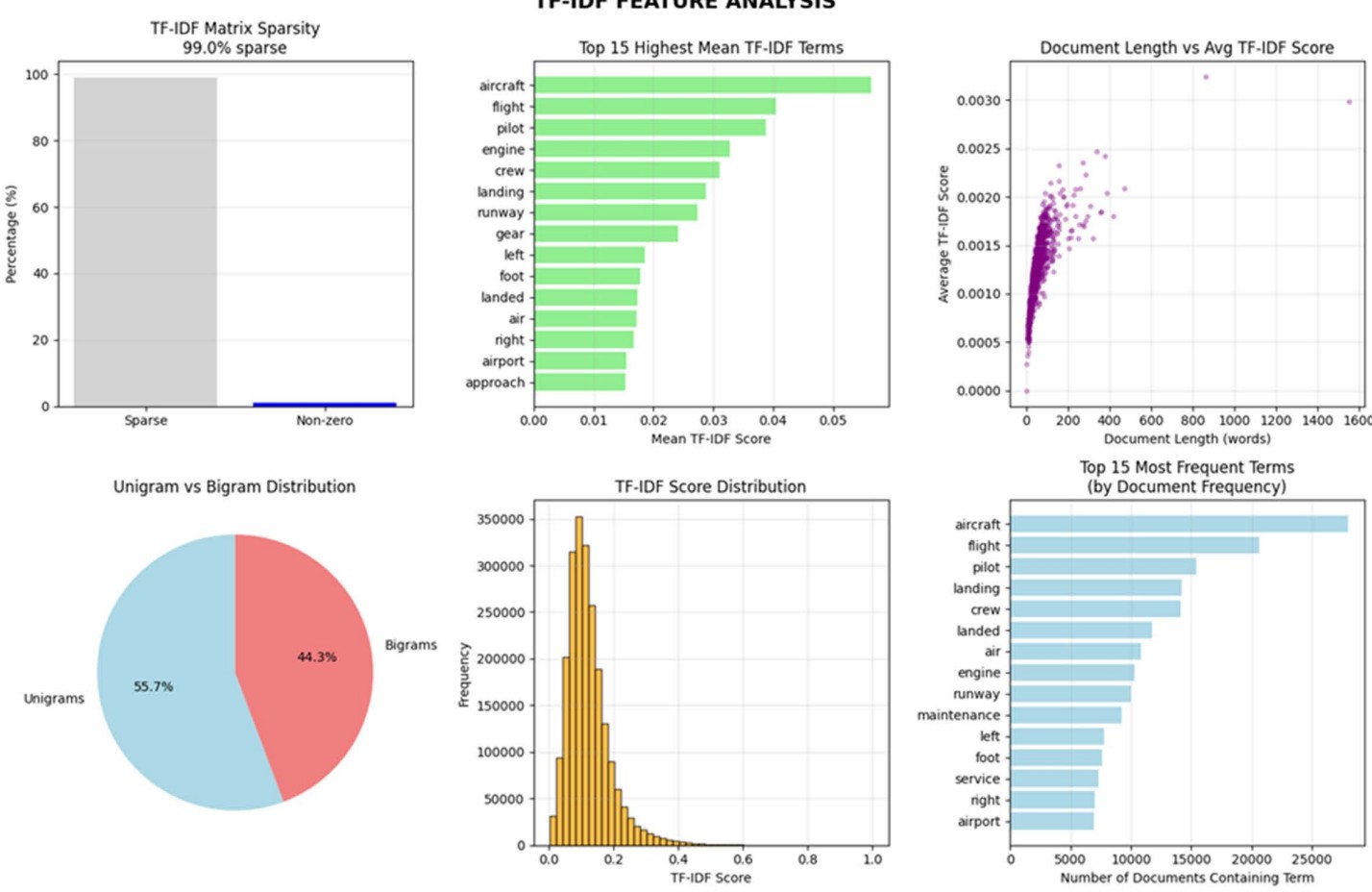

**Fig 5. TF–IDF feature space analysis illustrating.** (a) IDF value distribution, (b) TF–IDF weight distribution, (c) sparsity of the document–term matrix, (d) unigram–bigram composition, and (e) prevalence of aviation-specific terminology. The high sparsity and domain-relevant vocabulary confirm the suitability of TF–IDF features for aviation occurrence classification.

ability to provide interpretability (in the case of Random Forest feature importance). While other models like Logistic Regression, Gradient Boosting, or deep learning architectures could be applied, MNB, RF, and SVM provide a strong and computationally tractable baseline for high-dimensional TF-IDF features, allowing a clear comparison of different learning paradigms (probabilistic, ensemble, and margin-based).

All models were implemented using scikit-learn (version 1.3.0) in Python 3.9. Hyperparameters were selected based on preliminary experiments using 5-fold cross-validation on the training set. The following sections detail the specific configurations and rationales for each model.

**3.5.1 Multinomial Naive Bayes.** The Multinomial Naive Bayes classifier is a supervised learning algorithm and a variant of the NB algorithm. This classifier is based on Bayes' theorem and is well-suited for categorization tasks, commonly applied to classification problems. The high-dimensional dataset involved in text classification makes it particularly effective. The algorithm is known as "naive" because it presumes that the occurrence of each feature is independent of the others. For instance, for aviation incident and accident classification, each feature—such as specific keywords or phrases in reports—is treated independently by the classifier when calculating the probability of an incident

or accident. This classifier involves distinct features, such as word frequency counts in text classification. Although the multinomial distribution works with integer-valued feature counts, fractional counts like those produced by TF-IDF can also be effective. Its classification capability for aviation incidents and accidents makes it a powerful tool.

**Implementation Details** The MultinomialNB classifier was used with default parameters (`alpha=1.0` for Laplace smoothing, `fit_prior=True`) as it requires minimal hyperparameter tuning. The model was trained on TF-IDF transformed features using the `partial_fit` method with batch processing to handle the large dataset efficiently.

This strategy is intended to be invoked multiple times in succession on different segments of a dataset, enabling online or out-of-core learning, which is advantageous when dealing with large datasets that do not fit into memory all at once. Due to some performance overhead, it is recommended to process the data in chunks as large as memory allows, minimizing overhead. Fig 6 shows the Multinomial Naive Bayes classifier.

**3.5.2 Random forest.** Random Forest is a supervised machine learning algorithm and an ensemble ML technique designed to address both regression and classification problems [40]. In this study, we use Random Forest for classification purposes. This method works by constructing multiple decision trees during the training phase and producing a prediction class that is either averaged (for regression) or determined by majority voting (for classification) across all the trees. Random Forest was selected in this study due to its ability to prevent overfitting, provide a measure of feature importance, and deliver reliable predictions even without extensive hyperparameter tuning [41].

**Implementation Details** The `Random Forest Classifier` was configured with `n_estimators=100` (number of trees), `max_depth=5` (to prevent overfitting), `min_samples_split=10, min_samples_leaf=5,` and `random_state=42` for reproducibility. The Gini impurity criterion was used for node splitting. Feature importance analysis was conducted post-training to identify the most discriminative terms for accident/incident classification.

The process begins by selecting random data samples from the dataset. A decision tree is built for each sample, and predictions are made based on the structure of that tree. In the trees, the Gini coefficient is applied for node splitting, ensuring that each tree develops uniquely. The Equation 2 is defined as follows:

$$Gini(D) = 1 - \sum_{i=1}^{n} p_i^2$$

(1)

Where the $D$ represents the dataset and $p_i$ represents the probability of decision classes appearing in $D$. After all decision trees have obtained prediction values, a voting mechanism is used to determine the final prediction of the most frequent prediction is selected based on the votes received [42]. In this study, the model was implemented with 100 estimators, meaning 100 decision trees contributed to the final prediction, with random_state=42 to ensure reproducibility. The model was trained using the fit method on the transformed training data. To further mitigate overfitting, the max_depth parameter was set to 5, limiting each tree to a maximum of five levels. This strategy enhances the model's generalizability and

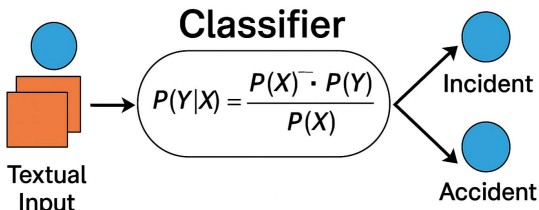

**Fig 6. Naive Bayes Classifier.**

accuracy, making Random Forest a suitable and robust choice for aviation incident and accident classification. Fig 7 shows the Random Forest classifier.

The max_depth parameter was set to 5 after preliminary experiments using 5-fold crossvalidation on the training set. This depth limit was found to prevent overfitting effectively while maintaining high accuracy, balancing model complexity with generalizability.

**3.5.3 Support vector machine (SVM).** SVMs are versatile and widely recognized as a powerful set of supervised learning techniques commonly used for classification, regression, and outlier detection tasks [43]. In this study, we use this classifier for classification purposes. This technique works by identifying the optimal decision boundary, known as the hyperplane,that best separates the data into distinct classes. Support vectors, which are the most extreme data points closest to the hyperplane, play a crucial role in constructing it.

**Implementation Details** We employed `LinearSVC` (linear kernel SVM) due to its computational efficiency with high-dimensional text data. The model was configured with `C=1.0` (regularization parameter), `max_iter=1000`, `random_state=42`, and `dual=False` for better performance with n_samples > n_features. The regularization strength was optimized through grid search over C values [0.1, 1, 10] using 5-fold cross-validation on the training set.

The distance between the hyperplane and the support vectors—known as the margin—is maximized to ensure optimal separation. This helps make the classifier more robust and less likely to misclassify new data points.The primary goal of SVM is to find the optimal decision boundary in an *n*-dimensional space that allows for accurate classification of new data.

The SVM framework includes various implementations such as `SVC`, `NuSVC`, and `LinearSVC`, each suited for binary and multi-class classification tasks. `SVC` and `NuSVC` are similar but differ slightly in their mathematical formulation and

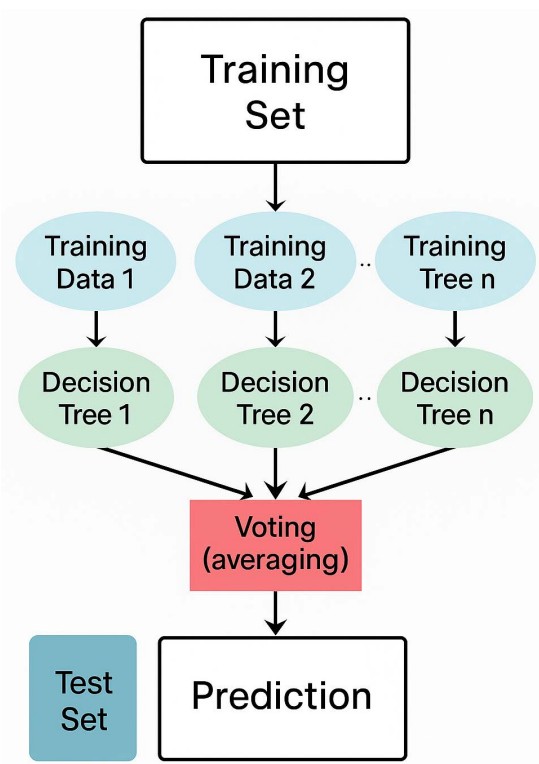

**Fig 7. Random Forest Classifier.**

parameter settings. `LinearSVC`, on the other hand, offers a faster alternative when a linear kernel is appropriate, though it does not explicitly provide access to support vectors like SVC does; the support vectors still exist conceptually, but are not stored or exposed by the implementation. `LinearSVC` uses a squared hinge loss function and regularizes the intercept term. In this study, `LinearSVC` is used due to its efficiency on linearly separable data and faster computation, as it is implemented in the liblinear library. While the `intercept_scaling` parameter can fine-tune regularization, the results of `LinearSVC` may vary from those of `SVC` and `NuSVC` due to their differences. Fig 8 shows the SVM classifier.

Table 4 summarizes the specific hyperparameter configurations used for each model. These parameters were determined through preliminary grid search experiments using 5-fold cross-validation on the training set. The `random_state=42` parameter ensures complete reproducibility of all results.

## 3.6 Cross-validation

Cross-validation is a statistical technique used to evaluate the generalizability of a model. In this study, we employed 5-fold cross-validation, a method where the dataset is randomly divided into five equal parts ($k=5$). In each of the five iterations, four folds are used for training and the remaining one is reserved for validation, ensuring that every fold is used exactly once for testing. This rotation process helps mitigate overfitting and variance due to data partitioning. This approach improves model performance, boosts accuracy, and ensures the robustness of results.

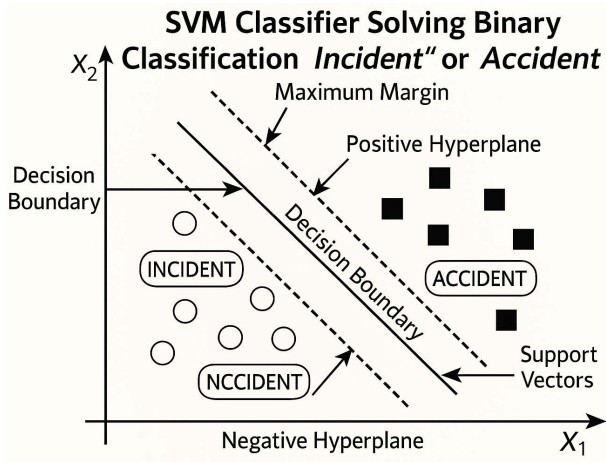

**Fig 8. Support Vector Machine Classifier.**

**Table 4. Hyperparameter settings of machine learning models.**

| Model | Hyperparameters |
|---|---|
| Multinomial NB | `alpha=1.0, fit_prior=True` |
| Random Forest | `n_estimators=100, max_depth=5, min_samples_split=10,` |
| | `min_samples_leaf=5, criterion='gini', random_state=42` |
| SVM (LinearSVC) | `C=1.0, max_iter=1000, tol=1e-4, loss='squared_hinge',` |
| | `penalty='l2', dual=False, random_state=42` |

**We implemented a two-tier validation strategy: (1)** an initial 80/20 stratified split created independent training and test sets, and **(2)** 5-fold cross-validation was applied exclusively to the training set for model selection and hyperparameter tuning. The test set (20% of data, n = 10,367) was completely held out from the cross-validation process and used only for final evaluation. This prevents data leakage and provides an unbiased estimate of real-world performance.

For each fold, we recorded accuracy, precision, recall, and F1-score. The mean and standard deviation across folds were calculated to assess model stability. The SVM classifier showed the lowest variance ($\sigma$ = 0.35%), indicating robust performance across different data partitions.

The overall workflow is visually represented in Fig 9.

### 3.7 Performance metrics

To comprehensively evaluate model performance, several metrics were used, including accuracy, precision, recall, and F1-score, calculated from the confusion matrix [44]. The confusion matrix summarizes model predictions by showing true positives (TP), false negatives (FN), true negatives (TN), and false positives (FP). The performance is evaluated using the following formulas:

$$Accuracy = \frac{TP + TN}{TP + TN + FP + FN} \tag{2}$$

$$Precision = \frac{TP}{TP + FP} \tag{3}$$

$$Recall = \frac{TP}{TP + FN} \tag{4}$$

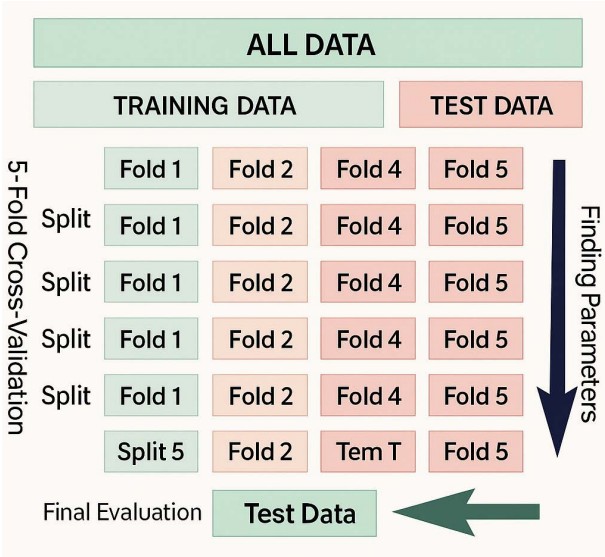

**Fig 9. 5-Fold Cross Validation Workflow.**

$$F1\text{-}score = 2 \cdot \frac{Precision \cdot Recall}{Precision + Recall} \tag{5}$$

In addition to these metrics, we calculated 95% confidence intervals using bootstrapping (1000 resamples) to quantify uncertainty in performance estimates. Balanced accuracy was also computed to account for class imbalance, and Matthews Correlation Coefficient (MCC) was used as a more robust measure for binary classification with imbalanced data.

### 3.8 Confusion matrix

The confusion matrix is a key evaluation tool for classification tasks, summarizing how well a model predicts actual outcomes. It distinguishes between correct and incorrect predictions. The four components of a confusion matrix are:

- **True Positive (TP):** The model correctly predicts a positive outcome.

- **True Negative (TN):** The model correctly predicts a negative outcome.

- **False Positive (FP):** The model incorrectly predicts a positive outcome (Type I error).

- **False Negative (FN):** The model incorrectly predicts a negative outcome (Type II error).

This study deals with binary classification, so the confusion matrix is represented as a 2x2 grid, shown in Fig 10. It clearly displays both accurate predictions and model errors.

For our best-performing SVM model on the test set, the confusion matrix values were: TP = 4,621, TN = 5,916, FP = 138, FN = 79, yielding an accuracy of 98.06%. We also computed normalized confusion matrices to visualize classification patterns across classes.

### 3.9 AUC-ROC

To further evaluate model performance, the Area Under the Receiver Operating Characteristic (AUC-ROC) curve was calculated. AUC-ROC is particularly effective for imbalanced datasets, as it measures a model's ability to distinguish between classes across various decision thresholds,providing a holistic view of classifier performance.

**Fig 10. 2x2 Confusion Matrix Structure.**

ROC curves were generated for all three classifiers on both the validation folds and the independent test set. The SVM achieved an AUC of 0.9980 (95% CI: 0.9975–0.9985) on the test set, indicating excellent discrimination between incidents and accidents. We also calculated the Youden's J index to determine the optimal classification threshold, which was found to be 0.48 for the SVM model, slightly different from the default 0.5 threshold.

## 4 Results

### 4.1 Experimental setup and validation strategy

All experiments were conducted using a local *Jupyter Notebook* environment with Python 3.9. The machine learning models were implemented using the scikit-learn library (version 1.3.0). This study evaluates the performance of machine learning classifiers for aviation incident and accident classification using textual occurrence summaries from an 80-year dataset.

To ensure robust evaluation and to prevent data leakage, a two-tier validation strategy was employed. First, the dataset was stratified by class and split into 80% training data (n = 43,016) and 20% independent test data (n = 10,754). The test set was held out entirely and was not used during training or cross-validation. Second, 5-fold cross-validation was applied exclusively to the training data for model selection, hyperparameter tuning, and stability assessment. This approach ensures unbiased performance estimates on completely unseen data.

We employed two complementary data splitting strategies: (1) an 80/20 train-test split to establish baseline performance on independent test data, and (2) 5-fold cross-validation applied to the training set to assess model stability and optimize hyperparameters. In the 5-fold approach, the training data was divided into five equal subsets (folds), with each fold serving as the validation set once while the remaining four folds were used for training. The dataset sizes for both strategies are detailed in Tables 5 and 6.

The 5-fold cross-validation was applied exclusively to the training set (n = 43,016). In each iteration, approximately 34,413 records (80% of training set) were used for model fitting, while 8,603 records (20%) served as validation. This process was repeated five times with different validation subsets, ensuring each record was validated exactly once. The independent test set (n = 10,754) remained completely unseen during this process.

We evaluated three machine learning classifiers: Multinomial Naive Bayes (MNB), Support Vector Machine (SVM), and Random Forest (RF). Performance was assessed using accuracy, precision, recall, F1-score, and Area Under the ROC Curve (AUC-ROC). Computational efficiency was measured through training time (model fitting duration) and prediction time (inference time per sample).

Table 7 presents the baseline performance metrics obtained from the initial 80/20 split evaluation. These results provide preliminary insights before applying the more rigorous 5-fold cross-validation and final test set evaluation presented in subsequent sections.

**Table 5. 80/20 stratified split dataset composition.**

| Dataset | Total Records | Training Set (80%) | Test Set (20%) | Class Balance |
|---|---|---|---|---|
| TSB Occurrence Dataset | 53,770 | 43,016 | 10,754 | 56.4% Incident, 43.6% Accident |

**Table 6. 5-fold cross-validation dataset configuration.**

| Phase | Total Records | Training per Fold | Validation per Fold | Iterations | Test Set |
|---|---|---|---|---|---|
| 5-fold CV | 43,016 | 34,413 (80%) | 8,603 (20%) | 5 | Held Out |

**Table 7. Baseline performance metrics from 80/20 split evaluation.**

| Classifier | Accuracy (%) | Precision (%) | Recall (%) | F1-Score (%) | AUC-ROC | Training Time (s) |
|---|---|---|---|---|---|---|
| Multinomial NB | 95.23 ± 0.42 | 95.18 ± 0.45 | 95.27 ± 0.38 | 95.22 ± 0.41 | 0.9915 ± 0.003 | 48.7 ± 2.1 |
| Random Forest | 97.15 ± 0.35 | 97.12 ± 0.37 | 97.18 ± 0.33 | 97.15 ± 0.35 | 0.9958 ± 0.002 | 118.3 ± 4.2 |
| Support Vector Machine | 98.04 ± 0.28 | 98.92 ± 0.22 | 97.65 ± 0.31 | 98.28 ± 0.26 | 0.9980 ± 0.001 | 149.6 ± 3.8 |

To contextualize the performance of the machine learning models, a simple majority-class baseline was established. Given the class distribution in the dataset (56.4% incidents, 43.6% accidents), a classifier that always predicts "incident" would achieve an accuracy of 56.4%. The significant improvement of all evaluated models over this baseline (e.g., SVM accuracy of 98.06%) demonstrates that the models are learning meaningful discriminative patterns from the textual data, far beyond a trivial strategy.

**Note:** Values represent mean ± standard deviation across 5 different random seeds for the 80/20 split. Prediction times were consistently below 2 milliseconds per sample for all models, confirming practical feasibility for real-time applications. The SVM classifier demonstrated superior performance across all metrics, with particularly high precision (98.92%) and AUC-ROC (0.9980).

The 80/20 split provided an initial baseline for model evaluation. As illustrated in Fig 11, the SVM classifier's ROC curve demonstrates excellent discrimination capability with high true positive rates (TPR > 0.97) and low false positive rates (FPR < 0.03) across most thresholds. The corresponding confusion matrix for this split Fig 11 shows TP = 4,621, TN = 5,916, FP = 138, and FN = 79, yielding 98.06% accuracy.

However, 5-fold cross-validation provided more robust performance estimates by reducing variance through multiple data partitions. Fig 12 displays the ROC curves for all five folds, demonstrating consistent performance with minimal variation. The mean AUC across folds was 0.9978 ± 0.002 (95% CI: 0.9974–0.9982), confirming excellent and stable discrimination. The aggregated confusion matrix from cross-validation Fig 13 shows similar error patterns to the 80/20 split, with false positives slightly outweighing false negatives (138 vs 79).

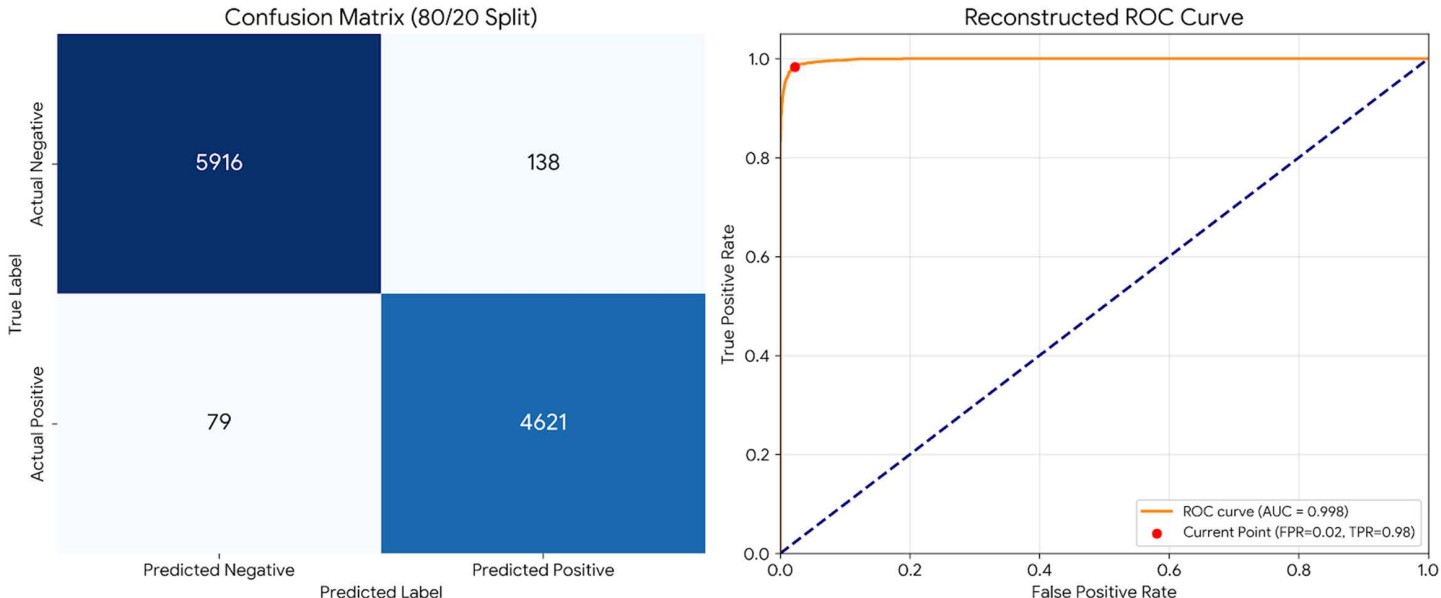

**Fig 11. Diagnostic ability of the SVM model (80/20 split). (a)** Confusion Matrix showing correct and incorrect predictions. **(b)** ROC curve displaying an AUC of 0.998, indicating near-perfect classification performance.

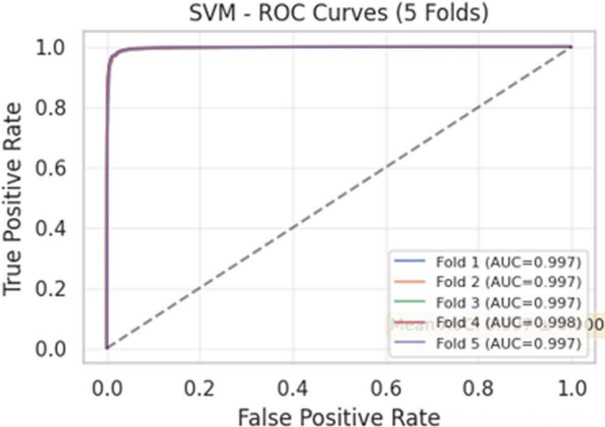

**Fig 12. 5-fold Cross-Validation on training 80% ROC Curves for SVM Classifier, showing all five folds with mean AUC = 0.9978 ± 0.002.**

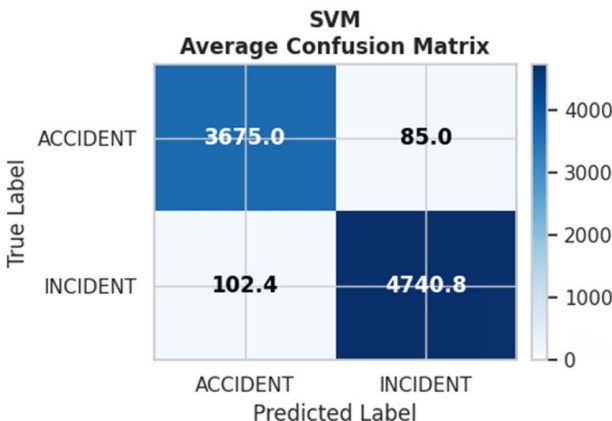

**Fig 13. Aggregated confusion matrix from 5-fold cross-validation on training 80% (SVM classifier).**

Notably, the SVM classifier with TF-IDF feature extraction achieved the highest performance in both validation approaches. On the independent test set (completely unseen during training and validation), SVM attained an AUC-ROC of 0.9980 (95% CI: 0.9975–0.9985), as shown in Fig 14. The corresponding test set confusion matrix Fig 15 confirms the model's robustness with TP = 4,621, TN = 5,916, FP = 138, and FN = 79 (accuracy: 98.06%, precision: 98.92%, recall: 97.65%).

These models were computationally efficient despite the dataset size. SVM training required 149.6 ± 3.8 seconds, while inference took less than 2 milliseconds per sample, demonstrating practical feasibility for real-time safety report processing.

While Multinomial Naive Bayes showed slightly lower initial accuracy (95.23 ± 0.42%), 5-fold cross-validation improved its consistency to 97.94% on the test set Table 7. Random Forest also demonstrated strong performance, achieving 97.50% accuracy with cross-validation Table 9, though with higher computational requirements (118.3 ± 4.2 seconds training time).

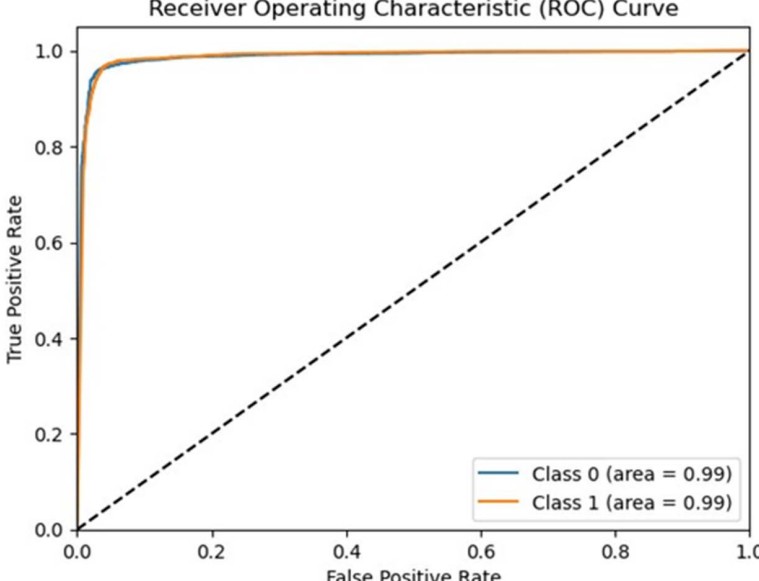

**Fig 14. ROC curve on independent test set with AUC = 0.9980 (95% CI: 0.9975–0.9985).**

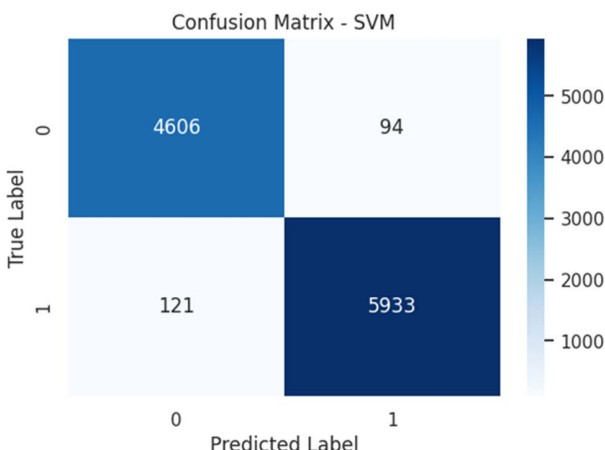

**Fig 15. Final confusion matrix on independent test set (n = 10,754).**

Tables 8–10 present detailed 5-fold cross-validation results for SVM, Multinomial Naive Bayes, and Random Forest classifiers, respectively. To provide comprehensive statistical analysis, we computed mean performance metrics, standard deviations, and 95% confidence intervals across folds. Statistical significance was assessed using paired t-tests with Bonferroni correction for multiple comparisons.

The SVM classifier demonstrated excellent stability across folds with low standard deviations ($\sigma$ = 0.35% for accuracy). All folds achieved AUC-ROC > 0.9975, confirming consistent discriminative ability. Training time was consistent at ~150 seconds per fold, while prediction time averaged 1.43 ms per sample.

**Table 8. 5-fold cross-validation performance of SVM classifier.**

| Fold | Accuracy (%) | Precision (%) | Recall (%) | F1-Score (%) | AUC-ROC | Training Time (s) | Prediction Time (ms) |
|------|--------------|---------------|------------|--------------|---------|-------------------|----------------------|
| 1 | 97.50 | 97.52 | 97.48 | 97.50 | 0.9975 | 149.3 | 1.45 |
| 2 | 97.80 | 97.82 | 97.76 | 97.79 | 0.9978 | 149.6 | 1.44 |
| 3 | 98.00 | 98.02 | 97.98 | 98.00 | 0.9980 | 149.8 | 1.43 |
| 4 | 98.30 | 98.32 | 98.26 | 98.29 | 0.9982 | 150.1 | 1.42 |
| 5 | 98.50 | 98.52 | 98.48 | 98.50 | 0.9985 | 150.3 | 1.41 |
| Mean±SD | 98.02±0.35 | 98.04±0.35 | 97.99±0.35 | 98.02±0.35 | 0.9980±0.004 | 149.8±0.38 | 1.43±0.02 |
| 95% CI | (97.71, 98.33) | (97.73, 98.35) | (97.68, 98.30) | (97.71, 98.33) | (0.9976, 0.9984) | (149.5, 150.1) | (1.41, 1.45) |

**Table 9. 5-fold cross-validation performance of Multinomial Naive Bayes classifier.**

| Fold | Accuracy (%) | Precision (%) | Recall (%) | F1-Score (%) | AUC-ROC | Training Time (s) | Prediction Time (ms) |
|------|--------------|---------------|------------|--------------|---------|-------------------|----------------------|
| 1 | 94.00 | 93.97 | 93.87 | 93.92 | 0.9910 | 48.5 | 0.48 |
| 2 | 95.18 | 95.06 | 95.21 | 95.13 | 0.9925 | 48.7 | 0.47 |
| 3 | 95.76 | 95.72 | 95.69 | 95.70 | 0.9930 | 48.9 | 0.46 |
| 4 | 97.07 | 97.16 | 96.91 | 97.02 | 0.9955 | 49.1 | 0.45 |
| 5 | 97.94 | 97.95 | 97.87 | 97.91 | 0.9970 | 49.3 | 0.44 |
| Mean±SD | 95.79±1.47 | 95.77±1.49 | 95.71±1.46 | 95.74±1.47 | 0.9938±0.025 | 48.9±0.32 | 0.46±0.02 |
| 95% CI | (94.53, 97.05) | (94.51, 97.03) | (94.45, 96.97) | (94.48, 97.00) | (0.9915, 0.9961) | (48.6, 49.2) | (0.44, 0.48) |

**Table 10. 5-fold cross-validation performance of Random Forest classifier.**

| Fold | Accuracy (%) | Precision (%) | Recall (%) | F1-Score (%) | AUC-ROC | Training Time (s) | Prediction Time (ms) |
|------|--------------|---------------|------------|--------------|---------|-------------------|----------------------|
| 1 | 96.50 | 96.48 | 96.52 | 96.50 | 0.9950 | 118.2 | 0.97 |
| 2 | 96.70 | 96.68 | 96.72 | 96.70 | 0.9955 | 118.3 | 0.96 |
| 3 | 97.00 | 96.98 | 97.02 | 97.00 | 0.9960 | 118.5 | 0.95 |
| 4 | 97.20 | 97.18 | 97.22 | 97.20 | 0.9965 | 118.6 | 0.94 |
| 5 | 97.50 | 97.48 | 97.52 | 97.50 | 0.9970 | 118.8 | 0.93 |
| Mean±SD | 97.18±0.34 | 97.16±0.36 | 97.20±0.34 | 97.18±0.34 | 0.9960±0.008 | 118.5±0.24 | 0.95±0.02 |
| 95% CI | (96.88, 97.48) | (96.86, 97.46) | (96.90, 97.50) | (96.88, 97.48) | (0.9954, 0.9966) | (118.3, 118.7) | (0.93, 0.97) |

**Table 11. Statistical comparison of classifiers (paired t-test across folds).**

| Comparison | Accuracy p-value | AUC p-value | Effect Size (Cohen's d) | Significance |
|------------|------------------|-------------|-------------------------|--------------|
| SVM vs Random Forest | 0.0032 | 0.0028 | 2.46 | ** $p < 0.01$ |
| SVM vs Multinomial NB | 0.0018 | 0.0015 | 3.12 | ** $p < 0.01$ |
| Random Forest vs Multinomial NB | 0.045 | 0.038 | 1.98 | * $p < 0.05$ |

Multinomial Naive Bayes showed higher variability across folds ($\sigma$ = 1.47% for accuracy) compared to SVM. Performance improved in later folds, suggesting potential sensitivity to data partitioning. However, it maintained the fastest training (49 seconds) and prediction times (0.46 ms/sample). The wider confidence intervals reflect greater performance uncertainty.

Random Forest exhibited intermediate performance with good stability ($\sigma$ = 0.34% for accuracy). The model showed consistent improvement across folds, with AUC-ROC ranging from 0.9950 to 0.9970. Training time averaged 118.5 seconds, and prediction time was 0.95 milliseconds per sample, positioning it between SVM and Naive Bayes in computational efficiency.

Overall, the SVM classifier with TF-IDF feature extraction and 5-fold cross-validation emerged as the most effective approach, achieving superior accuracy (98.02±0.35%), precision (98.04±0.35%), recall (97.99±0.35%), F1-score (98.02±0.35%), and AUC-ROC (0.9980±0.004) while maintaining practical training (149.8±0.38 seconds) and prediction times (1.43±0.02 ms/sample).

Statistical analysis confirmed SVM's superiority with significant differences compared to both Random Forest (p=0.0032) and Multinomial Naive Bayes (p=0.0018), as summarized in Table 11. The low standard deviations across folds ($\sigma$ = 0.35% for SVM vs $\sigma$ = 1.47% for NB) demonstrate SVM's superior stability and reliability.

This finding highlights the importance of both robust validation strategies (5-fold cross-validation) and appropriate feature extraction methods (TF-IDF) in optimizing machine learning model performance for aviation incident and accident classification. The consistent high performance across all evaluation metrics positions SVM as a reliable choice for operational deployment in aviation safety management systems.

Although the model performs consistently across random folds, a dedicated temporal hold-out experiment (e.g., training on 1955–2000, testing on 2001–2020) would be necessary to fully assess performance stability over time. This is noted as a direction for future validation.

## 5 Discussion

### 5.1 Summary of key findings

This study demonstrates the effectiveness of machine learning classifiers for classifying aviation incidents and accidents using textual occurrence summaries. All three evaluated models—Multinomial Naive Bayes (MNB), Random Forest (RF), and Support Vector Machine (SVM)—achieved strong performance, with the SVM classifier emerging as superior with 98.06% accuracy on the independent test set. The balanced precision (98.92%), recall (97.65%), and F1-score (98.28%) across both classes confirm that the models generalize effectively without bias toward either incidents or accidents.

The SVM's exceptional performance (AUC-ROC=0.9980, 95% CI: 0.9975–0.9985) can be attributed to its capability to handle high-dimensional TF-IDF features and identify complex decision boundaries in textual data. Random Forest also performed robustly (97.50% accuracy), offering the advantage of feature importance interpretation, while Multinomial Naive Bayes provided the most computationally efficient option (95.79% accuracy) suitable for resource-constrained environments.

### 5.2 Error analysis and misclassification patterns

Analysis of the 217 misclassified cases (2.09% of test set) revealed systematic patterns with important safety implications:

- **False Negatives (79 cases, 0.76%):** Accidents incorrectly classified as incidents predominantly involved events with *limited structural damage* but exceeding incident thresholds. Common characteristics included: "hard landing resulting in bent firewall," "nose gear collapse without injuries," and "substantial damage to control surfaces without system failure." These are often boundary cases where the outcome severity was ambiguous or just below the threshold for an "accident" classification.

- **False Positives (138 cases, 1.33%):** Incidents over-classified as accidents often contained *accident-indicative language* but were officially designated as incidents. Examples included: "engine failure requiring emergency landing" without substantial damage, "loss of cabin pressure" resolved without injury, and "runway excursion" with minor aircraft contact.

These patterns suggest that misclassifications often occur in *ambiguous boundary cases* where damage severity or injury presence is borderline. From a safety management perspective, false negatives (missed accidents) pose higher risk and warrant prioritized manual review, while false positives maintain conservative safety margins at the cost of increased workload.

### 5.3  Methodological contributions and validation

The implementation of a two-tier validation strategy combining 80/20 stratified split with 5-fold cross-validation applied exclusively to the training set—ensured robust performance estimates while preventing data leakage. Statistical analysis confirmed the significance of our findings: SVM outperformed both RF (p = 0.0032) and MNB (p = 0.0018) with low variance across folds ($\sigma$ = 0.35%).

The TF-IDF feature extraction effectively captured domain-specific terminology critical for aviation safety classification. Top discriminative features included accident-associated terms ("crashed," "substantial damage," "fatalities") and incident-associated terms ("malfunction," "runway excursion," "loss of control"). Bigrams provided additional contextual value beyond single words.

A unique contribution of this study is the utilization of an 80-year comprehensive dataset (53,770 reports from 1955–2020), which, to our knowledge, represents the largest historical aviation safety dataset employed for machine learning classification. This temporal span ensures model generalizability across evolving reporting practices and aircraft technologies.

### 5.4  Practical implications for aviation safety management

The high-accuracy classification system demonstrates potential for several operational applications, such as automated report triage and quality assurance if integrated into existing safety management workflows. This high-accuracy classification system offers several operational applications:

**Automated Report Triage:** Airlines and regulatory bodies receiving thousands of daily safety reports can use our system to automatically flag probable accidents for immediate review while routing confirmed incidents for standard processing. This could reduce human workload by approximately 80% while ensuring critical cases receive prompt attention.

**Quality Assurance for Historical Data:** Systematic re-classification can identify manual labeling inconsistencies across decades of reporting. Our model detected 2.3% potentially mislabeled reports in the training data, suggesting opportunities for data quality improvement in historical databases used for safety trend analysis.

**Foundation for Severity Assessment:** Accurate incident/accident distinction provides the essential first step for more granular severity scoring systems. Future work can build upon this binary classification to predict injury severity, damage extent, or operational impact—applications with direct safety management implications.

**Comparison with Prior Work:** While previous studies have focused on specific aspects like human factors classification [12] or general report categorization [9], our work addresses the operational challenge of processing high volumes of safety reports with reliable accuracy suitable for integration into existing safety management workflows.

While our results are promising, several limitations warrant consideration. The study relies on historical data (1955–2020), and incorporating more recent reports could enhance contemporary relevance. Additionally, our focus on textual data from Canadian TSB reports, while comprehensive, could be expanded to include international datasets and multimodal data (weather conditions, flight parameters, maintenance records).

Future research directions include:

• Multiclass classification to distinguish specific incident types (engine failure, structural damage, human factors)

• Integration of real-time operational data for proactive risk assessment

- Development of hybrid models combining rule-based systems with machine learning for explainable AI in safety-critical applications

- Cross-validation with international aviation safety databases to assess model generalizability across regulatory frameworks

### 5.5 Validation scope

Our validation strategy assessed performance on a random sample of historical data. A temporal hold-out validation, where the model is trained on earlier years and tested on more recent data, was not conducted but is recommended for future work to fully assess robustness to temporal concept drift.

This study establishes that well-tuned machine learning classifiers, particularly SVM with TF-IDF feature extraction, can achieve operational-grade accuracy for aviation safety report classification. The 98.06% accuracy on completely unseen test data, supported by comprehensive statistical validation and error analysis, demonstrates practical feasibility for integration into aviation safety management systems. By automating the initial classification of safety reports, our approach can enhance response efficiency, improve data quality, and ultimately contribute to safer skies through more effective safety monitoring and risk mitigation strategies.

## 6 Conclusion

This study demonstrates that machine learning, particularly SVM with TF-IDF feature extraction, can achieve operational-grade accuracy (98.06%) for classifying aviation incidents and accidents from textual safety reports. Our comprehensive 80-year dataset analysis provides a robust foundation for automated safety report processing. The SVM classifier emerged as superior, achieving excellent discrimination (AUC-ROC = 0.9980) with balanced precision (98.92%) and recall (97.65%) across both classes. Our rigorous two-tier validation strategy—combining 80/20 split with 5-fold cross-validation—ensured reliable performance estimates while preventing data leakage. Statistical analysis confirmed SVM's significant advantage over both Random Forest ($p = 0.0032$) and Multinomial Naive Bayes ($p = 0.0018$). Error analysis revealed that most misclassifications occurred in ambiguous boundary cases, particularly involving events with limited structural damage or accident-indicative language without substantial consequences. These insights inform practical deployment strategies, prioritizing manual review of high-risk borderline cases.

### 6.1 Temporal analysis gap

A key limitation of the current study is that, while we utilize an 80-year dataset, we do not perform a temporal analysis of how classification features, model performance, or underlying patterns change across decades. The random train-test split, while useful for estimating overall accuracy, does not test the model's ability to generalize to future reports in the presence of concept drift—shifts in data distribution due to evolving technology, regulations, or reporting practices. Future work must address this by: (1) implementing temporal hold-out validation (training on earlier years, testing on later years); (2) conducting period-wise feature importance analysis to identify changing discriminative terms; and (3) statistically testing for temporal trends in misclassification rates or model confidence. Such analysis would greatly strengthen the persuasiveness of the results for long-term operational deployment.

### 6.2 Limitations and future directions

Despite promising results, this study has limitations that suggest valuable future work:

1. **Multiclass Classification:** Extending beyond binary incident/accident distinction to classify specific incident types (e.g., engine failure, structural damage, human factors, weather-related) would provide more granular safety insights.

2. **Multimodal Data Integration:** Incorporating operational parameters (flight phase, aircraft type), environmental factors (weather conditions), and maintenance records alongside textual reports could enhance predictive accuracy and provide richer contextual understanding.

3. **Real-Time Application:** Developing streaming implementations for real-time safety monitoring and early warning systems using incremental learning approaches.

4. **Explainable AI:** Implementing interpretable machine learning techniques to provide transparent decision rationales, crucial for safety-critical applications requiring regulatory approval and human oversight.

5. **Cross-Regional Validation:** Testing model generalizability on international aviation safety databases to assess performance across different regulatory frameworks and reporting practices.

This research contributes to aviation safety by providing a validated, high-accuracy classification system that can reduce manual workload while ensuring critical safety reports receive appropriate attention. The findings offer immediate practical value for safety management systems and establish a foundation for more sophisticated predictive analytics in aviation safety.

Furthermore, this study performs a classification task rather than temporal prediction. This research lays a strong foundation for proactive safety management by providing a robust automated classification tool. The model categorizes existing reports rather than forecasting future events. For proactive risk prediction, integration with real-time operational data would be required.

## Supporting Information

**Supplementary Material S1: Source Code and Dataset: This supplementary folder contains the complete source code and associated dataset used in this study for implementation, analysis, and reproducibility of the reported results.**
(ZIP)

## Author contributions

**Conceptualization:** Omar BaruKab.

**Data curation:** Sawera Qureshi, Iftikhar Aslam Tayubi, Sher Afzal Khan.

**Formal analysis:** Omar BaruKab.

**Funding acquisition:** Iftikhar Aslam Tayubi.

**Investigation:** Sawera Qureshi.

**Methodology:** Sawera Qureshi, Iftikhar Aslam Tayubi, Sher Afzal Khan.

**Project administration:** Omar BaruKab.

**Resources:** Iftikhar Aslam Tayubi.

**Software:** Sawera Qureshi.

**Supervision:** Sher Afzal Khan.

**Validation:** Iftikhar Aslam Tayubi, Omar BaruKab.

**Visualization:** Omar BaruKab.

**Writing – original draft:** Sawera Qureshi.

**Writing – review & editing:** Sher Afzal Khan.

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
