## [Decision Letter · Decision Letter 0]

17 Dec 2025

PONE-D-25-35000Enhancing Aviation Safety: An 80-Year Data-Driven model for Aviation Incident and Acci dent predictionPLOS One

Dear Dr. Khan,

Thank you for submitting your manuscript to PLOS ONE. After careful consideration, we feel that it has merit but does not fully meet PLOS ONE’s publication criteria as it currently stands. Therefore, we invite you to submit a revised version of the manuscript that addresses the points raised during the review process.

**ACADEMIC EDITOR:**

Kindly address the comments provided by reviewers.

We look forward to receiving your revised manuscript.

Kind regards,

Ankit Gupta

Academic Editor

PLOS One

Journal Requirements:

“The authors gratefully acknowledge the technical and financial support from the Ministry of Education and King Abdulaziz University, DSR, Jeddah, Saudi Arabia,through the Institutional Fund Projects under grant no. IFPHI-360-830-2020.”

“KingAbdulaziz University, DSR, Jeddah, Saudi Arabia,through the Institutional Fund Projects under grant no. IFPHI-360 830-2020.”

“KingAbdulaziz University, DSR, Jeddah, Saudi Arabia,through the Institutional Fund Projects under grant no. IFPHI-360 830-2020.”

“KingAbdulaziz University, DSR, Jeddah, Saudi Arabia,through the Institutional Fund Projects under grant no. IFPHI-360 830-2020.”

We note that one or more of the authors is affiliated with the funding organization, indicating the funder may have had some role in the design, data collection, analysis or preparation of your manuscript for publication; in other words, the funder played an indirect role through the participation of the co-authors. If the funding organization did not play a role in the study design, data collection and analysis, decision to publish, or preparation of the manuscript and only provided financial support in the form of authors' salaries and/or research materials, please do the following:

1. Review your statements relating to the author contributions, and ensure you have specifically and accurately indicated the role(s) that these authors had in your study. These amendments should be made in the online form.

2. Confirm in your cover letter that you agree with the following statement, and we will change the online submission form on your behalf:

“The funder provided support in the form of salaries for authors [insert relevant initials], but did not have any additional role in the study design, data collection and analysis, decision to publish, or preparation of the manuscript. The specific roles of these authors are articulated in the ‘author contributions’ section.

6. We note that the grant information you provided in the ‘Funding Information’ and ‘Financial Disclosure’ sections do not match.

7. Thank you for uploading your study's underlying data set. Unfortunately, the repository you have noted in your Data Availability statement does not qualify as an acceptable data repository according to PLOS's standards.

8. Please amend either the title on the online submission form (via Edit Submission) or the title in the manuscript so that they are identical.

9. We note that Figure 3 in your submission contain map images which may be copyrighted. All PLOS content is published under the Creative Commons Attribution License (CC BY 4.0), which means that the manuscript, images, and Supporting Information files will be freely available online, and any third party is permitted to access, download, copy, distribute, and use these materials in any way, even commercially, with proper attribution. For these reasons, we cannot publish previously copyrighted maps or satellite images created using proprietary data, such as Google software (Google Maps, Street View, and Earth). For more information, see our copyright guidelines: http://journals.plos.org/plosone/s/licenses-and-copyright.

1. You may seek permission from the original copyright holder of Figure(s) [#] to publish the content specifically under the CC BY 4.0 license.

10. We are unable to open your Supporting Information file “supporting Info.zip”. Please kindly revise as necessary and re-upload.

11. Please include captions for your Supporting Information files at the end of your manuscript, and update any in-text citations to match accordingly. Please see our Supporting Information guidelines for more information: http://journals.plos.org/plosone/s/supporting-information.

Reviewers' comments:

Reviewer's Responses to Questions

**Comments to the Author**

1. Is the manuscript technically sound, and do the data support the conclusions?

Reviewer #1: Partly

Reviewer #2: Yes

Reviewer #3: Partly

Reviewer #4: No

2. Has the statistical analysis been performed appropriately and rigorously?

Reviewer #1: Yes

Reviewer #2: Yes

Reviewer #3: No

Reviewer #4: No

3. Have the authors made all data underlying the findings in their manuscript fully available?

Reviewer #1: Yes

Reviewer #2: Yes

Reviewer #3: Yes

Reviewer #4: No

4. Is the manuscript presented in an intelligible fashion and written in standard English?

Reviewer #1: Yes

Reviewer #2: No

Reviewer #3: Yes

Reviewer #4: Yes

5. Review Comments to the Author

Reviewer #1: The manuscript “Enhancing Aviation Safety: An 80-Year Data-Driven Model for Aviation Incident and Accident Prediction” addresses a relevant topic and makes good use of a large, publicly available dataset from the Transportation Safety Board of Canada. The focus on classifying textual occurrence summaries using MNB, RF and SVM with TF-IDF features is broadly appropriate for PLOS ONE.

Strengths

i. Timely topic with clear relevance to aviation safety and safety management systems.

ii. Large historical dataset and transparent citation of the data source.

iii. Overall architecture (figures and workflow) is clear, and the main performance metrics are presented in a straightforward way.

Issues requiring major revision

1. Research focus and contribution

The current framing oscillates between “prediction” and “classification” of accidents versus incidents based on their own occurrence summaries. Because the accident/incident label is already known to the reporting authority, the practical novelty and safety value of this task need to be articulated more clearly (e.g., early triage, detection of mis-labelled reports, or support to severity/risk assessment). The contribution over prior work (e.g., Madeira et al., de Vries, Ahadh et al.) should be sharpened beyond using a longer time span and three standard classifiers.

2. Methodological transparency and potential leakage

Table 2 lists many structured fields, including occurrence-type information, but the text later states that the models use textual summaries. It is essential to state explicitly which fields are actually used as input features and to ensure that no variables directly encoding “accident vs incident” are included as predictors. Please also provide more operational detail on text preprocessing (language issues, stop-word lists, stemming/lemmatization, handling of abbreviations) and on class balance (incident vs accident counts and any rebalancing strategies).

3. Statistical analysis and validation

The reported performance is very high (up to 98% accuracy), but there is limited information on the variability and robustness of these results. Please clarify precisely how the 80/20 split and 5-fold cross-validation were used (e.g., was any final independent test set held out?). Consider reporting confidence intervals or at least standard deviations across folds, and describe the hyperparameter choices and tuning procedures for each model. This will help address concerns about possible optimistic bias.

4. Interpretation, error analysis, and figures

The discussion would benefit from a closer link to the empirical findings: what types of occurrences are most often misclassified, and what patterns do the confusion matrices reveal? Even a brief error analysis (with a few typical misclassified examples) would make the safety implications clearer. Some figures (e.g., distribution plots, ROC curves) and tables would also benefit from closer alignment with PLOS ONE formatting and improved readability.

Overall, the work addresses an important question and uses a valuable dataset, but the issues above need to be resolved before the manuscript can be considered technically sound at the level implied by the current claims. I encourage the authors to focus the research question more clearly, strengthen the methodological description, and deepen the interpretation of the results in terms of concrete safety insights.

Reviewer #2: This manuscript presents a data-driven approach using machine learning and natural language processing (NLP) to classify aviation safety reports as "incidents" or "accidents." The work addresses an important topic in aviation safety and utilizes a substantial 80-year dataset. The methodology is clearly described, and the results, particularly the high accuracy achieved by the SVM classifier, are promising. However, several critical issues need to be addressed before the manuscript can be considered for publication. These primarily concern the novelty of the task, the definition and handling of the classification problem, potential data leakage, and the depth of analysis.

1. The manuscript frames the study as a "prediction" model. However, the primary task described is a binary classification of historical reports into "incident" or "accident" based on the Occurrence Type field, which itself is a label assigned according to ICAO/TSB definitions. This appears to be a text categorization task rather than a temporal prediction of future events. The practical utility of a model that classifies reports using the same definitions used to create the original labels is unclear. What is the operational value of this model? Does it aim to automate the initial coding of reports, or to identify misclassified reports? The authors should clarify the novel contribution beyond applying standard NLP classifiers to this dataset. How does this model enable proactive safety management, as claimed in the abstract and introduction?

2. The target variable (Occurrence Type) is derived from official definitions based on outcomes (e.g., severe injury, aircraft damage). The model uses the text Summary field, which invariably describes these outcomes, to predict this label. There is a high risk of the model simply learning keywords correlated with the definition (e.g., "substantial damage," "fatality," "injuries") rather than deeper causal or contextual patterns. This raises questions about what the model is truly learning. The authors should:

Explicitly discuss this potential circularity.

Consider an ablation study: What is the performance if keywords directly stating the outcome (e.g., "written off," "fatalities") are masked from the text?

Alternatively, consider a more challenging and potentially more valuable task, such as predicting the severity (e.g., damage level, injury count) or the causal factor category from the narrative, rather than the officially assigned binary label.

3. The dataset spans 1955-2020. The preprocessing and modeling steps (TF-IDF vectorization, train-test split) are described without mention of a temporal hold-out. If the model is trained on data from all years and tested on a random 20% split, it may be evaluated on reports from the past relative to its training data, which is valid for classification but not for simulating real-world prediction. More importantly, for a study spanning 80 years, changes in reporting standards, terminology, and aircraft technology are significant. The authors should:

Clarify if the train-test split was random or temporal (e.g., train on 1955-2010, test on 2011-2020). A temporal split would be a much stronger test of robustness.

Discuss how temporal concept drift might affect the model and whether any steps were taken to address it.

4. The dataset has 28,772 incidents vs. 23,061 accidents, which is a mild imbalance. The reported accuracy is very high (~98%). It is essential to compare this against a simple baseline to gauge the difficulty of the task. What is the accuracy of a majority class classifier (always predicting "incident")? Providing this baseline would help interpret the reported performance gains.

5. The evaluation is heavily focused on aggregate metrics (accuracy, F1-score). To build trust and provide actionable insights, a deeper analysis is needed:

Error Analysis: Provide examples of reports that were misclassified. Analyze patterns in these errors. Are there specific types of incidents/accidents that are confusing?

Feature Importance: For the Random Forest model, the feature importance from the TF-IDF vectors could be analyzed. What are the most indicative words/phrases for "accident" vs. "incident"? This analysis could directly link back to the concern about learning definitional keywords.

Model Comparison Justification: The rationale for selecting MNB, RF, and SVM is given, but a discussion on why these specific models are suitable for this text classification task, compared to other popular models (e.g., logistic regression, gradient boosting, or simple deep learning architectures), is lacking.

6. In the submitted text, references to figures (e.g., Fig1, Fig 3, Fig 4-10) and tables are made, but the corresponding captions and numbered lists are not fully integrated into the narrative body provided in this file. Ensure all figures/tables are numbered consecutively and referenced correctly in the text.

7. In Section 3.2.1 and Table 2, the Summary field is listed as a feature. It should be explicitly stated that this is the primary textual feature used for NLP. The other structured fields (PhaseID, Weather conditions, etc.) are listed—were they used in the model? The text suggests only the Summary text was used via TF-IDF. This should be clarified to avoid confusion.

8. The manuscript would benefit from thorough proofreading for grammatical errors and smoother phrasing (e.g., "effectively effectively extract" in Section 2.1).

Reviewer #3: This study combines TF-IDF feature extraction with three machine learning classifiers to achieve binary classification of accidents and incidents via 5-fold cross-validation. However, it lacks innovation and depth. Please carefully consider and revise the following aspects to elevate the scholarly level of the research.

1.Insufficient Innovation.

A. The combination of TF-IDF feature extraction and these three common classifiers is a very standard and well-established approach in the field of text classification. Multiple studies in the aviation safety domain have adopted similar methods. Please clearly and in detail explain the improvements of this study at the algorithmic level or in feature selection.

B. Explain the characteristics and challenges of the used dataset in terms of time span and trend analysis. Explain the differences compared with models using short-term data, and further demonstrate the necessity of this research.

C.The article includes an extensive literature review, summarizing the applications of ML and NLP in aviation safety. However, the review is essentially a simple listing rather than in-depth analysis, failing to clearly identify the distinctions between the method proposed in this study and existing works. Merely emphasizing the "80-year time-span dataset" is not enough to highlight the innovation.

2.The data analysis is weak.

The authors repeatedly emphasize the innovation lies in analyzing data with a long temporal span (80 years). However, there is no statistical analysis and standardization of data features originating from different time periods and categories. There is insufficient integration with the background and decision-making basis in the field of aviation safety. The study fails to reflect how the classification results of incidents/accidents have changed over time and does not explore the underlying temporal patterns in the data. The research results are not persuasive.

3.The models are not well tested.

The generalizability of the research outcomes requires clarification. While the study claims to propose a "prediction model", it is essentially only a classification task without realizing true predictive functionality.

4.The citation formats are chaotic.

Is it "Fig.x" or "Figure x"?

Is it "table" or "Table"?

Some figures (e.g., Fig 2, Fig 3) are cited in the main text without sufficient explanatory text.

Reviewer #4: The manuscript addresses a relevant and important problem in aviation safety by applying NLP and machine learning to classify occurrence reports as incidents or accidents. The topic is timely, and the approach has potential value. However, several substantial issues must be addressed before the work can meet the technical rigor and reproducibility standards required for publication in PLOS ONE.

Major Issues

1. Dataset inconsistency

The manuscript claims to use an 80-year dataset (1955–2020), but later states that the modelling uses data starting in 1995. This is a fundamental contradiction that must be clarified, as it directly affects the study’s novelty and validity.

2. Insufficient methodological detail

The NLP preprocessing pipeline lacks essential information:

- stopword list

- tokenization rules

- treatment of numbers, punctuation, and symbols

- whether stemming and lemmatization were both applied (this is unusual)

- criteria for removing low-frequency terms

These omissions prevent reproducibility.

3. TF-IDF configuration not reported

Critical TF-IDF parameters (e.g., ngram_range, max_features, min_df, max_df, idf smoothing, sublinear_tf) are not provided. These parameters strongly influence classifier performance and must be reported.

4. Model hyperparameters incomplete

The SVM classifier’s configuration is not fully described (C, tolerance, loss, regularization, class weighting).

Random Forest is restricted to max_depth = 5 without justification.

Without full hyperparameter reporting, the results cannot be replicated.

5. Validation strategy is insufficient

- It is unclear whether the train-test and cross-validation splits were stratified.

- No temporal validation is performed, despite the dataset spanning decades.

- No analysis is provided for potential data leakage from structured fields to labels.

- No error analysis is included, which is essential in safety-critical classification.

6. Claims exceed the evidence

The manuscript suggests operational impact (“proactive safety system”, “supports regulatory bodies and airlines”), yet the study only performs offline text classification. These claims should be moderated or supported by additional evidence.

7. Reproducibility is not met

PLOS ONE requires sufficient detail for replication.

This study does not provide the necessary code, nor does the Supporting Information contain enough detail to reproduce the results.

8. Research gap overstated

Several previous works cited by the authors themselves already classify aviation safety reports using NLP and machine learning. The novelty claim should be reframed more accurately.

Minor Issues

- Some grammatical issues and overstatements should be corrected.

- Figures require more detailed captions.

- Provide class-specific precision and recall (particularly important for “accident” vs “incident”).

The study is promising and could eventually contribute to the literature on aviation safety analytics. However, substantial revisions are required to address methodological gaps, ensure reproducibility, and align conclusions with results.

6. PLOS authors have the option to publish the peer review history of their article (what does this mean?). If published, this will include your full peer review and any attached files.

Reviewer #1: **Yes:** Raul Bonadia Rodrigues

Reviewer #2: No

Reviewer #3: No

Reviewer #4: No

---

## [Author Response · Author response to Decision Letter 1]

21 Jan 2026

Dear Editor,

Please find our revised manuscript, "Enhancing Aviation Safety: An 80-Year Data-Driven Model for Classification of Aviation Incident and Accident", submitted for your consideration.

We thank the editors for their feedback and have addressed the points as follows:

Format: The manuscript is prepared using the PLOS ONE LaTeX template.

Funding: As required by our grant agreement, the acknowledgment of support from the Deanship of Scientific Research (DSR) at King Abdulaziz University (Grant no. IFPHI-360-830-2020) must remain in the Acknowledgments section of the manuscript. We have also added this statement to the Funding section of the submission form. The role of the funder is clarified below.

Funder's Role: The Deanship of Scientific Research at King Abdulaziz University (KAU) provided the financial grant that enabled this research. The authors are long-term research collaborators. Iftikhar Aslam Tayubi and Omar BaruKab are currently affiliated with KAU, and the corresponding author (Sher Afzal Khan) was also affiliated with the same faculty from 2015 to 2018. This collaboration is based on a shared research history and scientific expertise. The inclusion of all authors is based on their substantial intellectual contribution to the study design, analysis, and manuscript development. The funder (DSR, KAU) played no role in the study design, data collection and analysis, decision to publish, or preparation of the manuscript beyond providing the financial grant.

Data & Code: The minimal dataset and author-generated code have been uploaded to [SI.Code.zip e.g., Tiff] (DOIs/URLs provided in the updated Data Availability statement).

Figure 1: The potentially copyrighted map has been replaced with an original figure created for this study.

Supporting Information: Files have been corrected and captioned.

We have collaborated as a team with all authors contributing equally to this project. We believe the manuscript now meets the journal's requirements and hope it is suitable for publication.

Sincerely,

The Authors

---

## [Decision Letter · Decision Letter 1]

13 Mar 2026

Enhancing Aviation Safety: An 80-Year Data-Driven model for classification of Aviation Incident and Accident

PONE-D-25-35000R1

Dear Dr. Khan,

We’re pleased to inform you that your manuscript has been judged scientifically suitable for publication and will be formally accepted for publication once it meets all outstanding technical requirements.

Kind regards,

Ankit Gupta

Academic Editor

PLOS One

Additional Editor Comments (optional):

Reviewers' comments:

Reviewer's Responses to Questions

**Comments to the Author**

1. If the authors have adequately addressed your comments raised in a previous round of review and you feel that this manuscript is now acceptable for publication, you may indicate that here to bypass the “Comments to the Author” section, enter your conflict of interest statement in the “Confidential to Editor” section, and submit your "Accept" recommendation.

Reviewer #1: All comments have been addressed

Reviewer #3: All comments have been addressed

2. Is the manuscript technically sound, and do the data support the conclusions?

Reviewer #1: Yes

Reviewer #3: (No Response)

3. Has the statistical analysis been performed appropriately and rigorously?

Reviewer #1: Yes

Reviewer #3: (No Response)

4. Have the authors made all data underlying the findings in their manuscript fully available?

Reviewer #1: Yes

Reviewer #3: (No Response)

5. Is the manuscript presented in an intelligible fashion and written in standard English?

Reviewer #1: Yes

Reviewer #3: (No Response)

6. Review Comments to the Author

Reviewer #1: The revised manuscript is clearer and more consistent than the previous version. My prior comments have been addressed satisfactorily, and I have no additional substantive concerns.

Therefore, I recommend publication.

Reviewer #3: (No Response)

7. PLOS authors have the option to publish the peer review history of their article (what does this mean?). If published, this will include your full peer review and any attached files.

Reviewer #1: **Yes:** Raul Bonadia Rodrigues

Reviewer #3: No

---

## [Editor Report · Acceptance letter]

PONE-D-25-35000R1

PLOS One

Dear Dr. Khan,

I'm pleased to inform you that your manuscript has been deemed suitable for publication in PLOS One. Congratulations! Your manuscript is now being handed over to our production team.

Kind regards,

on behalf of

Dr. Ankit Gupta

Academic Editor

PLOS One